# A role for descending auditory cortical projections in songbird vocal learning

**Yael Mandelblat-Cerf\*†, Liora Las†, Natalia Denisenko†, Michale S Fee**

Department of Brain and Cognitive Sciences, McGovern Institute for Brain Research, Massachusetts Institute of Technology, Cambridge, United States

**Abstract** Many learned motor behaviors are acquired by comparing ongoing behavior with an internal representation of correct performance, rather than using an explicit external reward. For example, juvenile songbirds learn to sing by comparing their song with the memory of a tutor song. At present, the brain regions subserving song evaluation are not known. In this study, we report several findings suggesting that song evaluation involves an avian 'cortical' area previously shown to project to the dopaminergic midbrain and other downstream targets. We find that this ventral portion of the intermediate arcopallium (AIV) receives inputs from auditory cortical areas, and that lesions of AIV result in significant deficits in vocal learning. Additionally, AIV neurons exhibit fast responses to disruptive auditory feedback presented during singing, but not during nonsinging periods. Our findings suggest that auditory cortical areas may guide learning by transmitting song evaluation signals to the dopaminergic midbrain and/or other subcortical targets.

## Introduction

Most human behaviors, such as speech, music, or athletic performance, are learned through a gradual process of trial and error. In all of these behaviors, the motor actions are shaped during learning by an internal model of good performance (*Wolpert et al., 1995*). While a great deal has been learned about the neural mechanisms by which external rewards, such as food or juice drops, are represented in the brain and might shape future behavior (*Schultz et al., 1997*; *Hikosaka, 2007*), little is known about how the brain evaluates its own performance, where such self-evaluation is computed and how these signals might shape future behavior.

Here we use the songbird as a model system to understand how complex behaviors can be learned by self-evaluation of motor performance, rather than relying on explicit external rewards such as food or social reinforcement. Songbirds learn their vocalizations by storing a memory, or 'template', of their tutor's song (*Doupe and Kuhl, 1999*). By listening to themselves sing and comparing their own song to this template (*Konishi, 1965*), they gradually converge to what can be a nearly exact copy of the tutor's song. Vocal learning requires a basal-ganglia forebrain circuit called the anterior forebrain pathway (AFP), the output of which projects to cortical premotor vocal areas (*Nottebohm et al., 1976*; *Bottjer et al., 1984*; *Scharff and Nottebohm, 1991*). The AFP is crucial for driving learned changes in song and shaping plasticity in the song motor pathway (*Brainard and Doupe, 2000*; *Andalman and Fee, 2009*; *Warren et al., 2011*). A key component of this vocal learning circuit is Area X, a basal ganglia homologue containing both striatal and pallidal components (*Person et al., 2008*).

Of critical importance in understanding the mechanisms of vocal sensorimotor learning in the songbird is to determine where song is evaluated and how the resulting evaluation signals are transmitted to the AFP to shape learning. Some hypotheses have emphasized the possible role of HVC (used as a proper name), a premotor cortical circuit that receives auditory inputs and also projects to Area X (*Troyer and Doupe, 2000*; *Mooney, 2006*; *Prather et al., 2008*; *Roberts et al., 2012*). While the involvement of HVC in evaluating ongoing song remains an open question, several lines of evidence suggest that HVC does not transmit error-related signals to the AFP

**\*For correspondence:** ycerf@ bidmc.harvard.edu

†These authors contributed equally to this work

**Competing interests:** The authors declare that no competing interests exist.

**Reviewing editor**: Ronald L Calabrese, Emory University, United States

**eLife digest** Most new skills, from playing a sport to learning a language, are acquired through a gradual process of trial and error. While some of this learning is driven by direct external rewards, such as praise, much of it occurs when the individual compares their current performance with their own impression of what a 'correct' performance should be. The way that the brain responds to external rewards is relatively well understood, but much less is known about the processes used by the brain to evaluate its own performance.

One way to study this process is to examine how songbirds learn their songs. While in the nest, young male birds memorize another bird's song, usually that of their father. They learn to sing by comparing their own vocals with this memorized template, tweaking their song until the two versions match. Now, Mandelblat-Cerf et al. have identified a pathway in the brain that enables the birds to make this comparison and to use any discrepancies to improve their subsequent attempts.

Anatomical labeling experiments revealed that a brain structure called the arcopallium has a key role in this process. The ventral part of this structure (known as AIV) receives inputs from the auditory cortex—meaning that it has access to the bird's own song—and then forms connections with a specific group of neurons in the midbrain. These midbrain neurons, which communicate using the chemical transmitter dopamine, project to brain regions that ultimately control the movements involved in singing. This means that the AIV is ideally positioned to be able to evaluate and then adjust the song as required.

Consistent with this possibility, young zebra finches were less able to imitate a template song if their AIV was destroyed before they had started practicing. By contrast, destroying the AIV in adult birds who had already learned their song did not impair performance, indicating that the arcopallium circuit supports song learning rather than singing per se. Finally, recordings of neurons in the AIV made during singing revealed that this brain area sends signals about discrepancies between what the young bird tries to sing and what he hears himself sing.

In addition to providing further clues as to how the songbirds learn their songs, this work also highlights the fact that dopaminergic neurons in the midbrain—which are best known for being involved in our response to external rewards such as food and drugs—also contribute to learning that is driven internally.

(*Hessler and Doupe, 1999*; *Kozhevnikov and Fee, 2007*), or receive auditory inputs (*Hamaguchi et al., 2014*) during singing.

Another possibility is suggested by the large projection to Area X from dopaminergic nuclei in the midbrain—the ventral tegmental area (VTA) and substantia nigra pars compacta (SNc) (*Person et al., 2008*; *Figure 1A*). Inspired by the recently hypothesized role of dopaminergic signaling in reinforcement learning (*Houk et al., 1994*; *Schultz et al., 1997*; *Bayer and Glimcher, 2005*; *Tsai et al., 2009*), we and others have proposed that this dopaminergic input to Area X may play a role in song learning by signaling vocal errors (*Hara et al., 2007*; *Gale et al., 2008*; *Gale and Perkel, 2010*; *Fee and Goldberg, 2011*). Notably, a recent anatomical study in songbirds has described a projection to the dopaminergic midbrain from the arcopallium (*Gale et al., 2008*), an avian cortical region containing subtelencephalically projecting neurons analogous to those in deep layer-5 of mammalian cortex (*Dugas-Ford et al., 2012*; *Karten, 2013*).

Here we set out to examine the role of the cortical area projecting to VTA/SNc in song learning. First, we characterized the spatial extent of the arcopallial neurons projecting to VTA or SNc, and characterized the inputs to these neurons from auditory cortical fields. We then carried out lesions targeted to this area to examine the effect on tutor imitation and on vocal plasticity that occurs after deafening. Finally, we recorded from VTA/SNc-projecting neurons in the arcopallium of singing birds and observed fast transient responses to disruptive auditory feedback during singing. Altogether, our findings suggest that descending auditory cortical projections may play a role in song evaluation via projections to the dopaminergic midbrain and/or other subcortical targets.

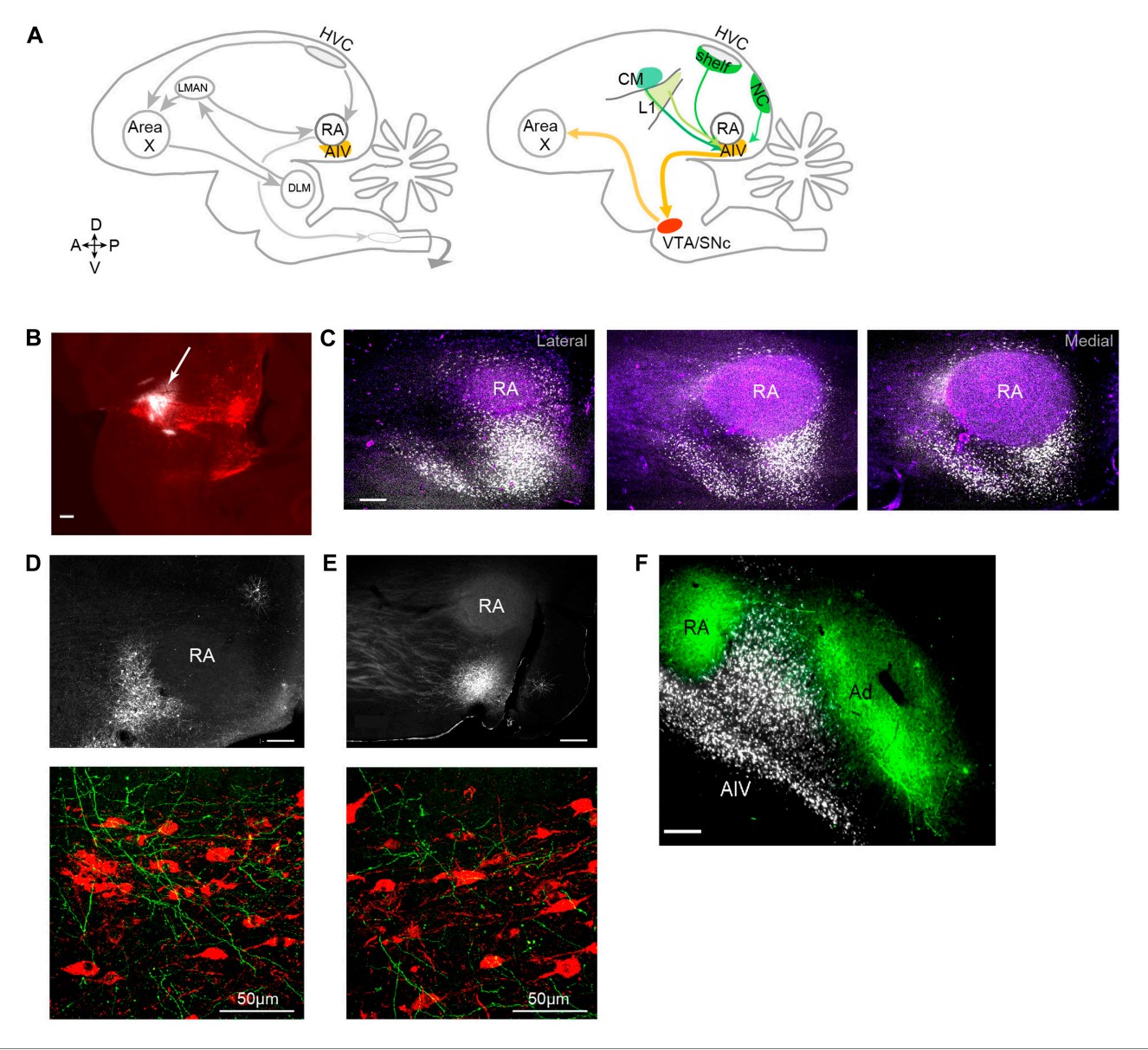

**Figure 1**. Characterization of avian 'cortical' areas projecting to the dopaminergic midbrain. (**A**) Left panel: schematic of the songbird brain (in sagittal view) showing classical song control brain areas. Nuclei HVC and RA form the descending song motor pathway, necessary for song production. Nucleus RA is in the arcopallium, a cortical-like region homologous to layer-5 neurons of the mammalian cortex. Also shown is the anterior forebrain pathway (AFP), a circuit necessary for vocal learning but not vocal production. The AFP consists of a basal ganglia homologue Area X, thalamic nucleus DLM, and cortical-like nucleus LMAN, which projects to RA. Right panel: schematic showing a set of pathways that are the focus of this paper, including a previously described projection to Area X from neurons in the dopaminergic midbrain nuclei VTA and SNc (*Gale et al., 2008*). Also shown is AIV, a part of the intermediate arcopallium found to project to VTA and SNc. Projections to AIV from auditory cortical areas L1, CM, and HVC-shelf are elucidated here. (**B**) Sagittal section showing the injection site of a retrograde tracer (CTB, white color, arrow) within VTA/SNc (TH-stained neurons, red). (**C**) Three sagittal sections through the arcopallium showing retrogradely labeled neurons in AIV (fluorescence, white) and relation to RA (dark field image, purple). Panel at left is most lateral; sections 200 μm apart. (**D**) Upper panel: image of injection site of a GFP-expressing virus (HSV, fluorescence white) in the anterior part of AIV, showing relation to RA (dark field image). Bottom panel: sagittal section of VTA/SNc showing anterogradely labeled fibers from AIV (green) and TH-stained neurons (red). (**E**) Same as **D**, with an injection site in AIV ventral to RA. All scale bars 200 μm unless indicated otherwise. In panels **A**–**E**, anterior is left; dorsal is up. (**F**) Coronal section showing axons in RA and Ad (green), anterogradely labeled from LMAN and LMAN-shell, respectively (Dextran 10 MW Alexa 488). AIV neurons (white) labeled by injection of a retrograde tracer (CTB) in VTA and SNc (medial, left; dorsal, up).

The following figure supplements are available for figure 1:

**Figure supplement 1**. Retrograde labeling of AIV neurons from VTA and SNc.

## Results

### An auditory cortical pathway to the dopaminergic midbrain

To elucidate the spatial pattern of cortical neurons projecting to the dopaminergic midbrain in song-birds, we made small injections of a retrograde tracer (cholera toxin subunit β, CTB) into VTA and SNc (n = 12 zebra finches, *Figure 1B*, see 'Materials and methods'). Consistent with an earlier report (*Gale et al., 2008*), we observed a distinct mass of retrogradely labeled cell bodies within the ventral-most extent of the intermediate arcopallium. Labeled neurons were distributed in a complex pattern that was highly consistent across birds (*Figure 1C*, *Figure 1—figure supplement 1*). The main cluster of labeled neurons was arranged in a stem-like column ventral to RA (robust nucleus of the arcopallium), while a distinct 'stripe' of labeled neurons was observed 300–400 μm anterior to the main cluster. A smaller number of labeled neurons formed a thin shell around the dorsal surface of RA, as previously described (*Gale et al., 2008*). We will refer specifically and collectively to the arcopallial areas containing neurons retrogradely labeled from the dopaminergic midbrain, described above, as the ventral intermediate arcopallium (AIV).

Injections of anterograde tracer (HSV viral vector expressing green fluorescent protein, GFP) within the anterior and posterior clusters of AIV neurons each revealed axonal arborization intermingled with TH-positive neurons in both VTA and SNc (*Figure 1D,E*, n = 6 birds).

It is important to elucidate the anatomical relation between the area we identify as AIV and the dorsal arcopallium (Ad), which has previously been implicated in vocal learning (*Bottjer and Altenau, 2010*), and also in locomotory control (*Feenders et al., 2008*; *Jarvis et al., 2013*) (called lateral intermediate arcopallium, LAI, in the latter studies). Ad was labeled by injections of anterograde tracer (Alexa 488 conjugated dextran 10 MW) into the nidopallium lateral to LMAN (LMAN-shell), and AIV was retrogradely labeled in the same birds by injection of CTB into VTA and SNc (n = 8 hemispheres). In coronally sectioned tissue, Ad was seen as a band of labeled fibers extending lateral to RA (*Figure 1F*), as previously described. Neurons retrogradely labeled from VTA/SNc were visible as a wedge of cell bodies ventrolateral to RA and ventromedial to Ad. Few labeled cell bodies were observed within the region of labeled fibers in Ad, although some retrogradely labeled neurons were seen in the thin 'neck' of labeled fibers connecting RA and Ad. Injections of retrograde tracer (CTB) into the anterior or posterior parts of AIV (n = 5) revealed no retrogradely labeled neurons in the nidopallium adjacent to LMAN (LMAN-shell). Thus, AIV and Ad appeared to be largely distinct non-overlapping regions.

To further elucidate the possible cortical afferents to AIV, we made injections of retrograde tracer (CTB) into different locations anterior and ventral to RA (n = 10 birds). Following injections into the anterior parts of AIV (n = 5 birds), retrogradely labeled neurons were reliably observed in auditory cortical areas HVC-shelf, caudal mesopallium (CM) and L1 (*Figure 2A–F*, *Figure 2—figure supplement 1A–C*) (*Kelley and Nottebohm, 1979*; *Vates et al., 1996*; *Mello et al., 1998*). We note that, while these previous studies have identified inputs to ventral arcopallium (RA-cup) from L1 and HVC-shelf, the projection described here from CM is novel. Following injections into the more posterior parts of AIV (n = 5 birds), retrogradely labeled neurons were observed in the caudal nidopallium (NC) posterior to HVC-shelf (*Figure 2G,H*).

To confirm that neurons in these four areas terminate within AIV, we carried out anterograde tracing using a GFP-expressing virus (HSV-GFP). Injections of anterograde tracer into the caudal nidopallium (NC) posterior to HVC-shelf revealed axonal arborization primarily in the posterior 'stem' region of AIV (*Figure 2—figure supplement 1G–H*, n = 8 birds). Injections of anterograde tracer into auditory cortical areas L1, CM, and HVC-shelf all revealed axonal arborization overlapping with neurons retrogradely labeled from VTA/SNc, particularly within the anterior 'stripe' region of AIV (*Figure 2I–L*, *Figure 2—figure supplement 1D–F*; L1 and CM, n = 4 birds each; HVC-shelf, n = 10 birds). In these experiments, axonal arbors were not restricted to the region of retrogradely labeled neurons in AIV, but were also seen anterior to the AIV 'stripe' and in the area directly anterior to RA.

The functional connectivity of projections from the auditory areas L1, CM, and HVC-shelf was further confirmed by electrically stimulating these areas and demonstrating short latency activation of multi-unit recording sites in AIV and of antidromically identified VTA/SNc-projecting AIV neurons (*Figure 3*). Together these findings suggest the existence of multiple descending pathways by which auditory information could reach the dopaminergic midbrain and other downstream auditory targets of the arcopallium (*Kelley and Nottebohm, 1979*; *Vates et al., 1996*; *Mello et al., 1998*).

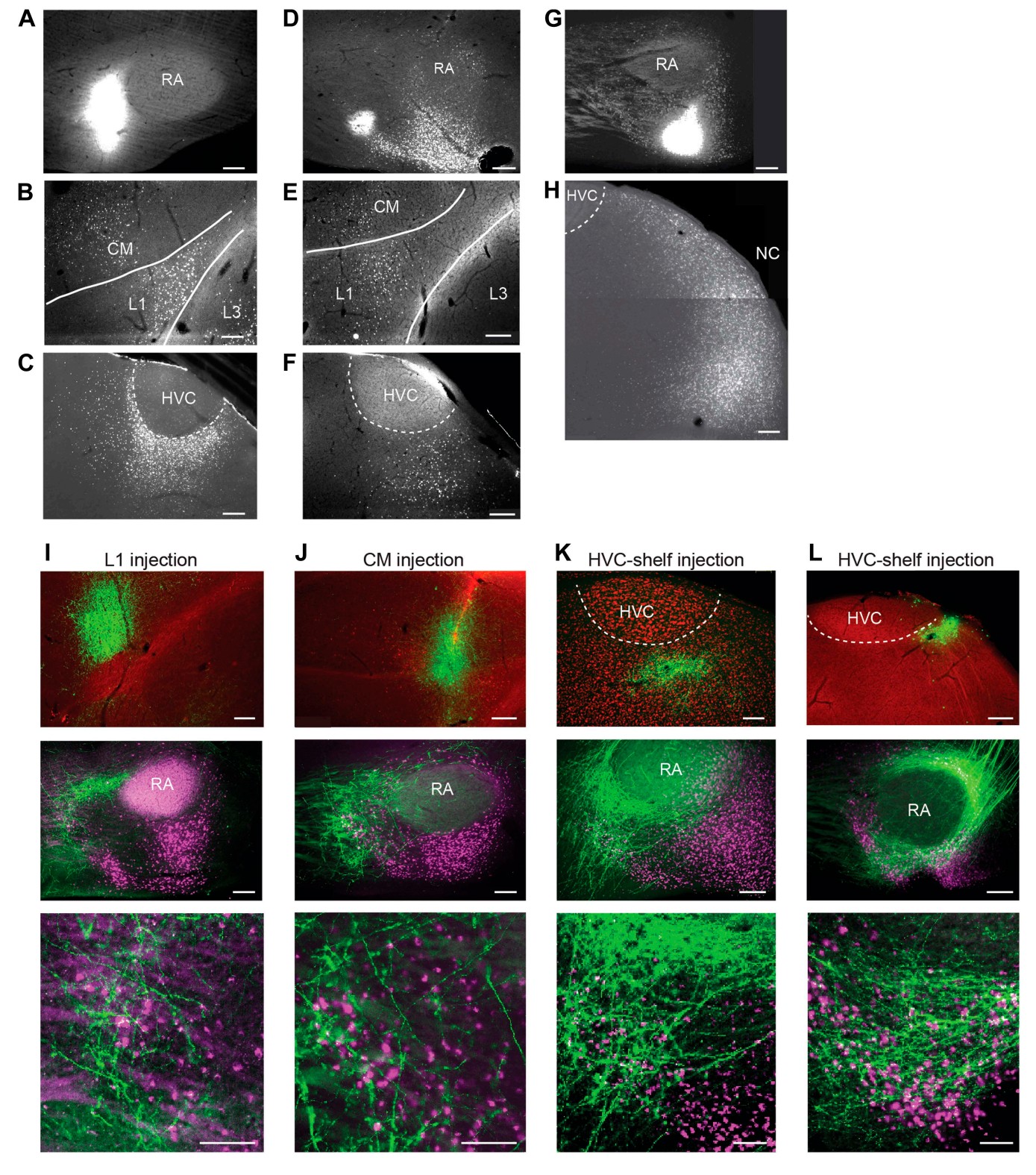

**Figure 2**. Cortical inputs to ventral intermediate arcopallium (AIV): retrograde and anterograde tracing. (**A**) Injection of a retrograde tracer (CTB) anterior to RA in the vicinity of AIV. (**B**) Retrograde label in cortical auditory areas L1 and CM resulting from the injection shown above. (**C**) Labeled neurons in HVC-shelf of the same bird. (**D–F**) Same as panels **A–C**. Injection of CTB was targeted to the anterior 'stripe' part of AIV. (**G**) Injection of CTB into the
*Figure 2. Continued on next page*

Figure 2. Continued

posterior part of AIV, directly ventral to RA. (**H**) Retrogradely labeled neurons in caudal nidopallium (NC) resulting from the injection shown above. (**I**–**L**) Anterograde tracing from auditory cortical areas L1, CM, and HVC-shelf. (**I**) Upper panel—injection of a GFP-expressing virus (HSV, green) into auditory field L1. Middle panel shows labeled axons in the vicinity of AIV neurons retrogradely labeled (purple) from VTA/SNc. Color in RA is due to auto fluorescence, not label. Bottom panel shows higher magnification view from image above. (**J**) Same as (**I**), but the injection of HSV-GFP was made into the auditory caudal mesopallium (CM). (**K**) Same as (**I**), but the injection of HSV-GFP was made into HVC-shelf. Scale bars: 100 µm for lower panels of **I**–**L**; 200 µm for all other panels.

The following figure supplements are available for figure 2:

**Figure supplement 1**. Cortical inputs to ventral intermediate arcopallium (AIV).

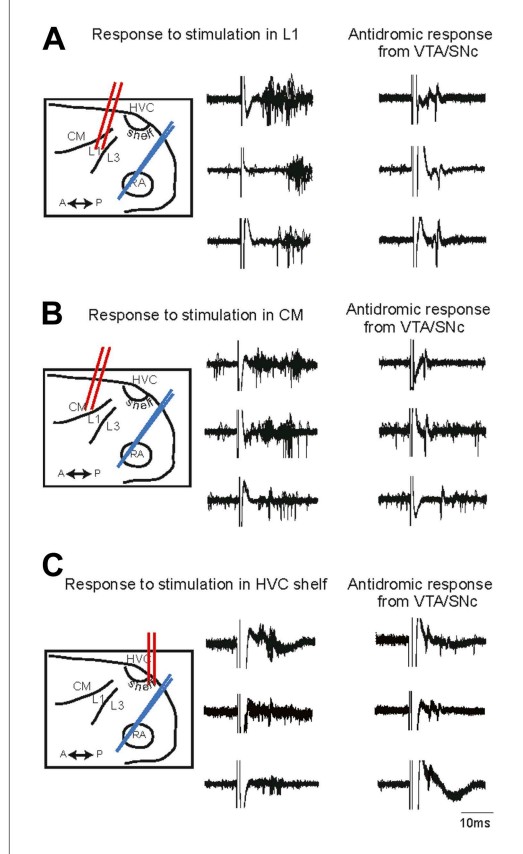

**Figure 3**. Electrophysiological verification of functional connectivity. Electrical stimulation in auditory cortical areas L1, CM, and HVC-shelf drives spiking in VTA/SNC-projecting neurons in AIV. (**A**) Schematic at left illustrates the location of the stimulating electrodes (red) in L1 and the location of the recording electrode (blue) in AIV. Middle traces: responses of 3 antidromically identified AIV neurons to L1 stimulation (overlayed responses from ten trials). Right traces: antidromic response of the same neurons to electrical stimulation in VTA/SNc. (**B** and **C**) The panels are analogous to those shown in (**A**), except the stimulation electrode is placed in caudal mesopallium (CM) or in the posterior part of HVC-shelf.

## Lesions of AIV disrupt song learning

To test the role of AIV in vocal learning and imitation, we carried out excitotoxic lesions targeted to AIV in young zebra finches, and examined the subsequent effect on song imitation. Birds were lesioned after being tutored in their home cage but prior to evidence of vocal imitation (***Figure 4A,B***, 48–51 days post hatch, dph). After the lesion surgery ('Materials and methods'), birds were maintained in isolation and their songs were recorded until adulthood (90 dph). Control siblings underwent the same tutoring protocol but did not receive AIV lesions. A quantification of song imitation (by analysis of pupil-tutor similarity, see 'Materials and methods') revealed that the AIV-lesioned birds exhibited deficits in vocal learning (***Figure 4C–F***). Lesioned birds had a significantly lower similarity to their tutor (imitation score) than their unlesioned control siblings (***Figure 4—figure supplement 1A***, paired *t* test, p=0.0048, rank-sum test between all controls and all AIV-lesioned birds p=0.0024). The overall distribution of acoustic features (***Tchernichovski et al., 2000***) was not significantly different between AIV-lesioned and control groups, suggesting that these lesions did not create gross abnormalities in song production (***Figure 4—figure supplement 1B–D***).

We investigated the possibility that the observed learning deficits after AIV lesions might be caused by the loss of neurons in dorsal arcopallium (Ad), which is dorsolaterally adjacent to AIV. In a subset of juvenile birds (n = 7), lesions were targeted to the portion of Ad most likely affected by our AIV lesion procedure (***Figure 5A***, 'Materials and methods'). These birds underwent the same tutor exposure and learning protocol as did the AIV-lesioned birds described above. The Ad-lesioned birds exhibited significantly higher song imitation scores compared to AIV-lesioned birds (Wilcoxon rank-sum test p=0.012), and no significant deficit in song imitation compared to non-lesioned controls (***Figure 5B***, Wilcoxon rank-sum test p=0.74). These birds exhibited no overt motor or locomotor deficits.

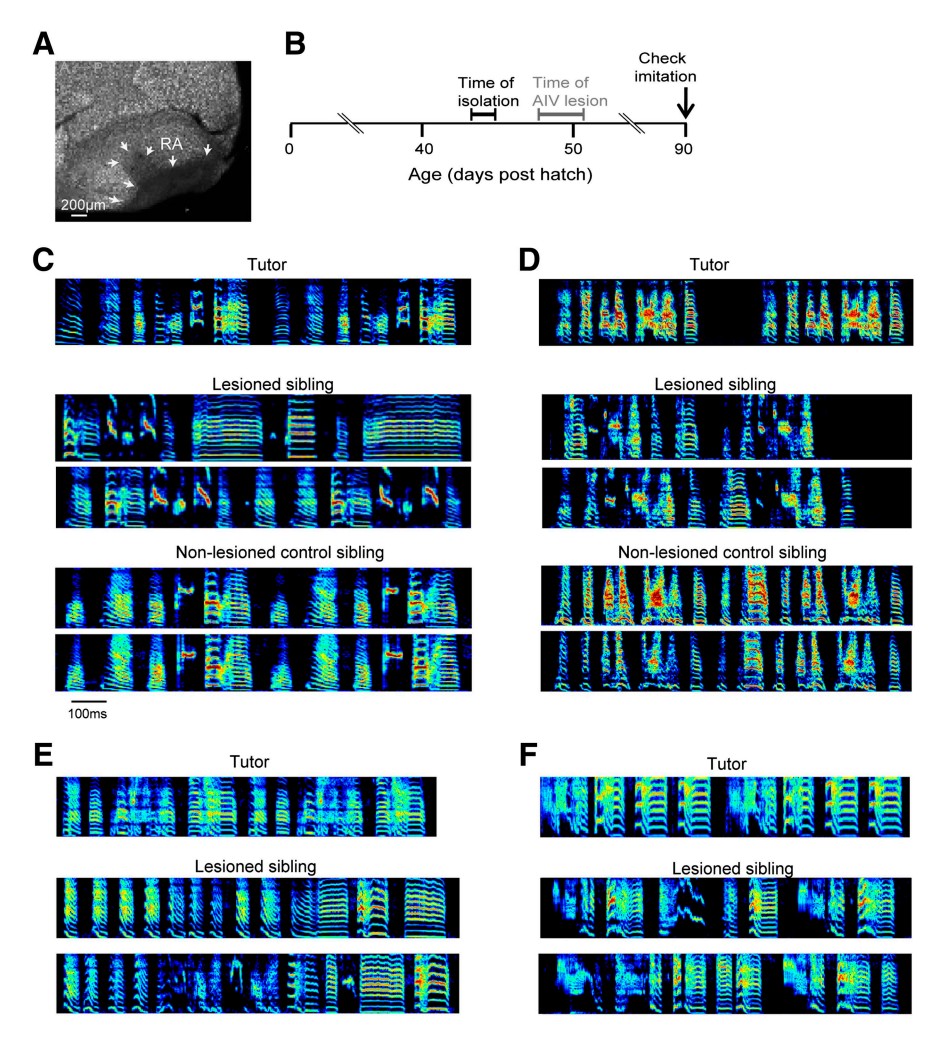

**Figure 4**. Bilateral lesions of AIV impair tutor imitation. (**A**) Excitotoxic lesion targeted to AIV by injection of NMA. Lesion borders indicated by arrowheads (sagittal section; anterior left, dorsal up). (**B**) Schematic timeline of experimental protocol. (**C**) Song spectrograms of an adult bird (90 dph) that underwent bilateral lesion of AIV as a juvenile, compared to an unlesioned control sibling. Top: spectrogram of tutor song. Middle: two example song spectrograms of the lesioned bird (45% lesion, 0.205 imitation score). Bottom: two example song spectrograms of the control sibling (0.235 imitation score). (**D**) Same as panel **C**, but for a different pair of birds (lesioned bird, 66% lesion, 0.1566 imitation score; control sibling, 0.27 imitation score). (**E** and **F**) Song spectrograms of two additional adult birds that underwent lesions of AIV as juveniles. These birds did not have control siblings. (Bird in panel **E**, 61% lesion, 0.097 imitation score; bird in panel **F**, 77% lesion, 0.114 imitation score.)
The following figure supplements are available for figure 4:

**Figure supplement 1**. Effect of AIV lesions on song motor production and imitation.

The lack of effect of Ad lesions, on either vocal learning or other aspects of motor behavior, led us to wonder whether complete lesions of Ad might have an observable effect. Larger lesions of Ad were carried out in three additional birds (the lesions were extended 0.2 mm more laterally, but within Ad). All three birds exhibited severe akinesia and immobility requiring the termination of the experiment. Because these birds did not sing, we were not able to assess the effects of the larger Ad lesions on vocal learning.

We wondered if the song imitation deficits produced by the AIV lesions described above were correlated with the size of the lesion. Several different lesion protocols were tested in this study, resulting

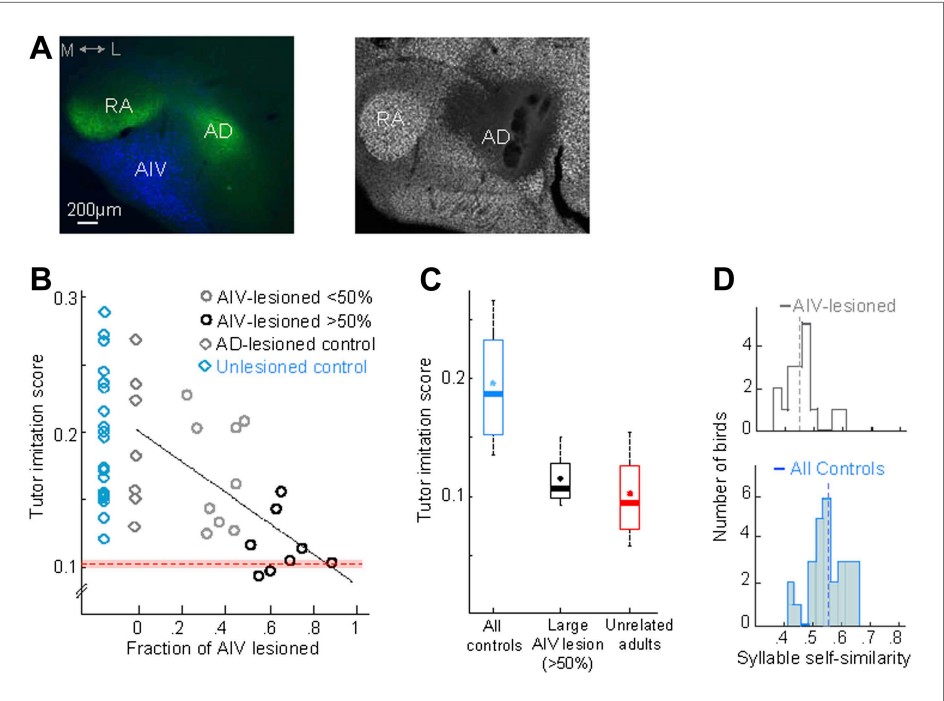

**Figure 5**. AIV lesions, but not Ad lesions, impair tutor imitation. (**A**) Left: coronal section showing the relation between AIV neurons (retrogradely labeled from VTA/SNc, blue) and nucleus Ad (anterograde labeling from LMAN-shell, green; medial, left; dorsal, up) Right: coronal section showing excitotoxic lesion of Ad, revealed by loss of NeuN staining (white, note that the lesion affected some of the overlying nidopallium). (**B**) Tutor imitation score plotted as a function of the extent of AIV lesion for each bird (hollow circles, n = 17 AIV-lesioned birds). Ad-lesioned birds (n = 7) are shown as 0% lesion (hollow gray diamonds) since Ad lesions had minimal impact on AIV. No significant impairment of song-imitation was observed in Ad-lesioned birds as compared to unlesioned controls (blue diamonds, n = 19 birds). Solid line denotes least square fit to Ad-lesioned and AIV lesioned data points. Red dashed horizontal line indicates mean of all similarity comparisons between 20 unrelated adult birds (red shaded area indicates SEM) (**C**) Boxplot of the distribution of imitation scores of all control birds (cyan, unlesioned, and Ad-lesioned controls) and birds with large AIV-lesions (black, >50% lesion). Also shown is a boxplot of the distribution of similarity scores of all pairwise comparisons between 20 unrelated adult birds (red). Whiskers denote 10–90 percentile. Asterisk in each boxplot denotes the mean, heavy line denotes median. (**D**) Distribution of syllable self-similarity in AIV-lesioned birds (top) and in control birds (bottom, unlesioned, and Ad-lesioned controls combined). Dashed lines denote mean of distributions.

The following figure supplements are available for figure 5:

**Figure supplement 1**. Effect of AIV lesions on song imitation, development of song stereotypy, and rate of singing.

in different extents of AIV lesion across the data set (n = 17 birds total). In all lesioned birds, the extent to which AIV was lesioned was quantified histologically at the age 90 dph ('Materials and methods'). Because Ad lesions minimally impacted AIV, Ad-lesioned birds (n = 7) were included in this analysis as sham lesions (0% AIV lesion). Song imitation scores were negatively correlated with the fraction of AIV that was lesioned (***Figure 5B***, linear regression using least square, R = 0.63, F-statistic p=0.0008, ***Figure 5—figure supplement 1A,B***).

Birds for which we estimated that more than half of AIV was lesioned (n = 8 birds, referred to as a large AIV lesion) produced poor imitations of the tutor song compared to controls (***Figure 5C***, rank-sum test, p=0.00016; Ad-lesioned and unlesioned control groups were pooled in this comparison). On average, birds with large AIV lesion exhibited an 87 ± 9% loss of imitation capacity compared to the controls ('Materials and methods'). The distribution of tutor imitation scores for birds with large lesions was not significantly different from the distribution of pairwise similarity scores between unrelated adult birds in our colony (***Figure 5C***, Wilcoxon rank-sum test, p=0.31). Altogether, our findings

suggest that the songs of birds with large AIV lesions were nearly as dissimilar to their own tutors as unrelated adult birds in our colony are to each other.

Importantly, the impaired vocal imitation in AIV-lesioned birds cannot be attributed to the amount of vocal practice. We compared the amount of singing in the period from surgery up to 90 dph for AIV-lesioned birds vs Ad-lesioned controls. No difference was observed between these groups (*Figure 5—figure supplement 1D,E*, *t* test, p=0.56). Not surprisingly, both Ad- and AIV-lesioned birds sang less than the unlesioned control group (ranksum p=0.003 and 0.016 respectively). However, within any experimental or control group, there was no correlation between the amount of singing and the degree of song imitation (F-statistics all comparisons, p>0.4).

In addition to impaired vocal imitation, AIV-lesioned birds developed adult songs that exhibited a significantly reduced overall song stereotypy (p=0.0045, rank-sum test, *Figure 5—figure supplement 1C*) (*Aronov et al., 2008*) and syllable stereotypy (p=0.01, rank-sum test, *Figure 5D*), compared to the control groups (Ad-lesioned and unlesioned control groups were individually significant, but were pooled in the comparisons stated above). Another common feature of the songs of AIV-lesioned birds was the presence of a large number of 'atypical' syllables that were uncharacteristically long in duration or contained patterns of acoustic modulation not usually observed in zebra finch song.

## AIV lesions do not have the same effect as deafening

Auditory feedback is necessary for vocal learning in juvenile birds, as revealed by experiments showing that birds deafened as juveniles are unable to learn normal songs (*Konishi, 1965*). Because AIV is a downstream target of auditory brain areas, we wondered if the effects of AIV lesions might be similar to deafening. We took advantage of the fact that deafening results in a dramatic degradation of song in young adults that have largely finished learning their songs (*Nordeen and Nordeen, 1992*; *Lombardino and Nottebohm, 2000*). For example, deafening at the age of 75–80 dph resulted in a near complete loss of song structure, typically within 1 week (n = 14 birds, *Figure 6A,D*, see 'Materials and methods' for quantification). In contrast to the effect of deafening, bilateral lesions targeted to AIV (n = 6 birds, 75–80 dph) produced no significant song degradation within a 2-week period following the lesion (*t* test p=0.28, *Figure 6B,D*).

Notably, AIV lesions had no immediate effect on song structure or song variability (paired *t* test p=0.52, *Figure 6—figure supplement 1A,B*), suggesting that AIV plays no direct role in song production or in the generation of vocal variability. These results also suggest that our AIV lesions did not significantly disrupt fibers of passage, either afferent fibers entering RA laterally from LMAN (lateral magnocellular nucleus of the anterior nidopallium, an area which conveys variability into the motor output during singing [*Kao et al., 2005*; *Ölveczky et al., 2005*]), or efferent fibers exiting RA rostrally (*Figure 6—figure supplement 1C–F*).

## AIV lesions inhibit deafening-induced song degradation

It has been shown that lesions of the basal ganglia–forebrain learning circuit, AFP, largely prevent the degradation of song that occurs after deafening, leading to the view that such degradation is an active process requiring vocal learning circuitry (*Brainard and Doupe, 2000*; *Nordeen and Nordeen, 2010*). Since we have hypothesized that AIV is involved in vocal learning, we wondered whether lesions targeted to this area might slow the degradation of song after deafening. To test this idea, we carried out bilateral cochlear removal in combination with bilateral AIV lesions (n = 15, 75–80 dph). Songs were recorded for 2 weeks following this procedure and were analyzed for similarity to the pre-surgery song. Song degradation after deafening was significantly slower in AIV-lesioned birds than in birds with intact AIV (*Figure 6C,D*, comparison at 7 days and 14 days post-lesion, *t* test p=0.0074, 0.0096 respectively). The reduced rate of deafening-induced song degradation after AIV lesion suggests a role for descending auditory cortical pathways, including possibly those projecting to the dopaminergic midbrain, in mediating the plastic changes in the AFP and motor pathways that occurs after deafening.

## AIV neurons respond to disruptive auditory feedback during singing

We next wanted to test if neurons in AIV—specifically those projecting to VTA/SNc—exhibit auditory responses during singing consistent with their hypothesized role in vocal learning. To address this

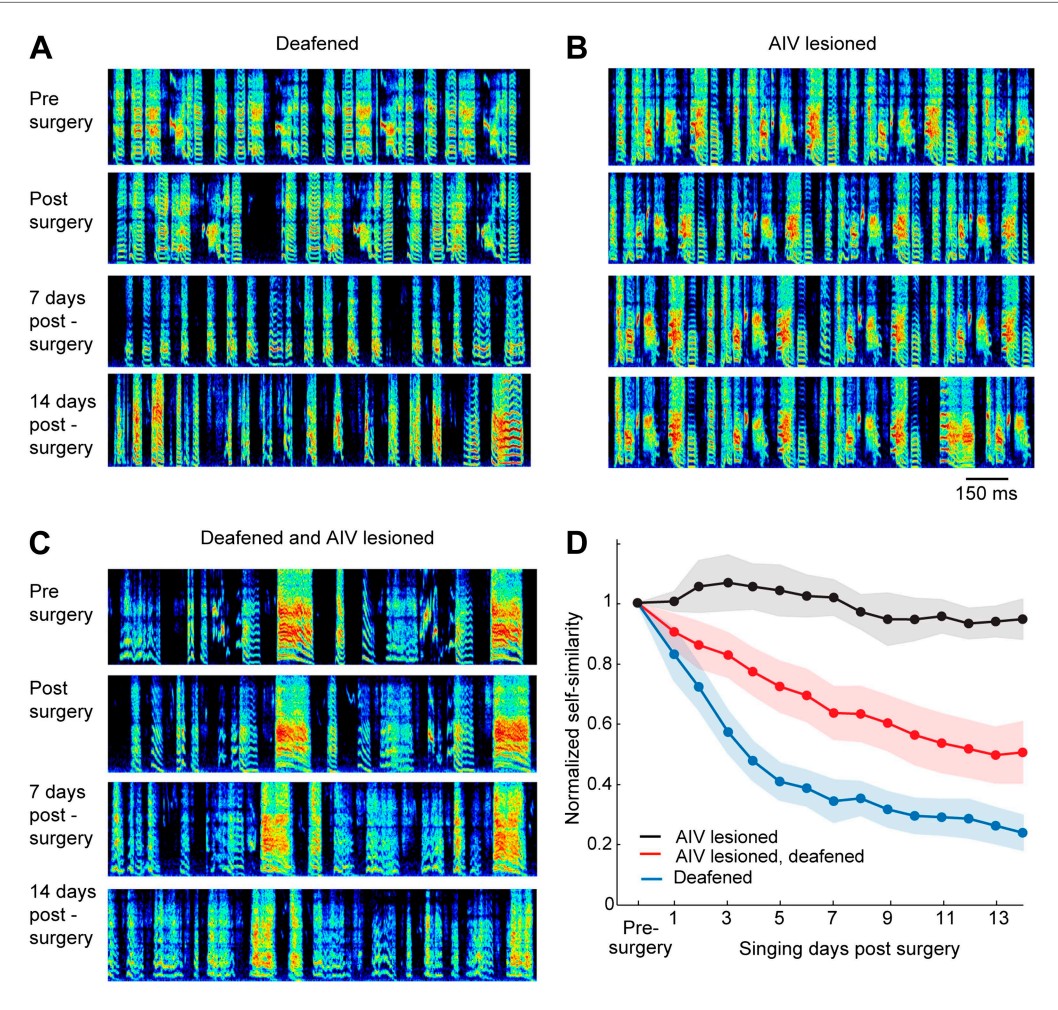

**Figure 6**. Effect of AIV lesion on adult song and on song degradation after deafening. (**A**) Examples of song spectrograms from a bird deafened at the age 80 dph. Shown from top to bottom; song before surgery (deafening), first song post-surgery, song 1 week post-surgery, and song 2 weeks post-surgery. Note rapid degradation of song structure within 1 week after deafening. (**B**) Examples of song spectrograms from a bird that underwent complete bilateral lesion of AIV at the age 80 dph. Note the lack of song degradation. (**C**) Song spectrograms from a bird that underwent both bilateral lesion of AIV and deafening at the age 80 dph. (**D**) Plot of song self-similarity, normalized to the average self-similarity of pre-surgery song. Note that lesioned birds exhibited a reduced rate of song degradation after deafening, compared to deafening alone, 1 week and 2 weeks post-surgery (*t* test, p<0.01).

The following figure supplements are available for figure 6:

**Figure supplement 1**. Immediate effects on song of excitotoxic and electrolytic lesions in AIV.

question we used a motorized microdrive to record neural activity in AIV of singing zebra finches (n = 7 birds, 90-120 dph). Recordings were targeted based on electrophysiological mapping of RA borders ('Materials and methods'). Recordings were made from 37 single units, of which 17 were antidromically identified and collision-tested VTA/SNc projectors (*Figure 7A,B*). We also recorded 13 multiunit sites exhibiting robust antidromic response to VTA/SNc stimulation ('Materials and methods', antidromic responses to VTA/SNc stimulation were restricted to the borders of AIV as defined by retrograde labeling, *Figure 7—figure supplement 1*). AIV single-units discharged at low rates during singing (1–10 Hz, *Figure 7C*), and exhibited only small changes in average firing rate compared to non-singing (*Figure 7D*, significant firing rate change in 15/37 neurons, paired *t* test p<0.05). Only a small fraction of AIV neurons exhibited significant firing rate modulations locked to the song motif (n = 4/37).

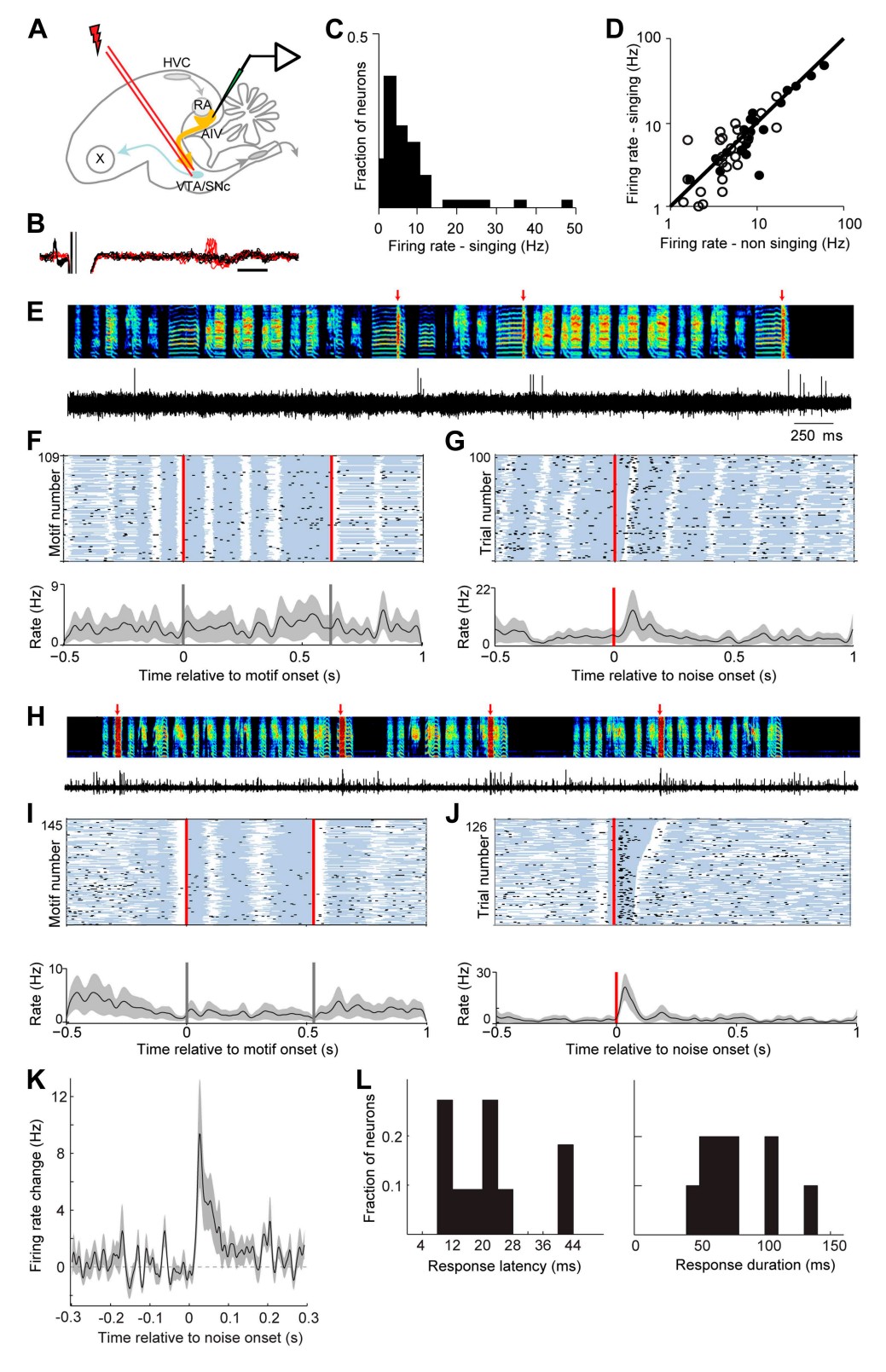

**Figure 7**. AIV neurons projecting to VTA/SNc exhibit error-related auditory responses during singing. (**A**) Schematic diagram of recording setup showing stimulating electrode placed in VTA/SNc for antidromic identification of AIV neurons. (**B**) Voltage traces showing an antidromically evoked spike of a VTA/SNc-projecting
*Figure 7. Continued on next page*

*Figure 7. Continued*

AIV neuron (red spikes). Collision test shown in black traces. (**C**) Distribution of firing rates of AIV neurons during singing. (**D**) Firing rates of AIV neurons during singing vs non-singing (antidromically identified neurons, hollow circles; non-identified AIV neurons, filled circles). (**E**) Recording of an antidromically identified AIV neuron during singing (song spectrogram, top; extracellular voltage trace, bottom) with presentation of noise bursts (red arrows). (**F**) Motif-aligned spike raster plot of the AIV neuron from panel **E** during singing with no noise bursts presented. Each row in the raster plot corresponds to a rendition of the song motif. Each dot corresponds to a spike. Along each row, gray areas denote syllables and white areas denote silent gaps. Two vertical red lines denote motif onset and offset. Shaded area along the histogram denotes SEM. Note lack of singing-related firing rate modulations. (**G**) Spike raster plot and spike histogram aligned to noise bursts during singing. Red vertical line denotes noise onset. Other notations are as in **F**. Note the brief response to the noise burst during singing. (**H–J**) Another example of an antidromically identified AIV neuron (larger spike waveform) recorded during singing. (**I**) Motif-aligned raster plot of the AIV neuron from panel **H**. (**J**) Spike raster plot and histogram aligned to noise bursts during singing. Rasters are sorted by the duration of the syllable in which the noise burst occurred. Note that the response to noise burst occurs during different syllable types. (**K**) Average peri-stimulus histogram for all antidromically identified AIV neurons, aligned to noise burst onset during singing. (**L**) Distribution of response latency after noise burst (left) and response duration (right), for AIV neurons that showed a significant response to noise bursts.

The following figure supplements are available for figure 7:

**Figure supplement 1**. Simultaneous mapping of VTA/SNc-projecting neurons in AIV by antidromic activation and retrograde tracing.

These modulations did not appear to be premotor in nature, and were substantially weaker than those reported in premotor song control nuclei (*Leonardo and Fee, 2005*), or auditory cortical areas (*Sen et al., 2001*).

While natural song learning proceeds slowly (*Tchernichovski et al., 2001*), playback of disruptive auditory feedback, such as brief bursts of broadband noise, can induce rapid learned changes in song structure over the course of a few hours (*Tumer and Brainard, 2007*; *Andalman and Fee, 2009*). Such learning has been interpreted as evidence that distorted auditory feedback can be used to introduce experimentally controlled 'errors' in song performance (*Tumer and Brainard, 2007*; *Andalman and Fee, 2009*; *Fee and Goldberg, 2011*). More than 40% of the VTA/SNc-projecting neurons exhibited a significant neuronal response to distorted auditory feedback (50 ms noise bursts) presented during singing (*Figure 7E–K*, $p < 0.02$ for n = 7/17 neurons, comparison of spike count in a 150 ms window before and after noise burst onset for each neuron; paired $t$ test, $p < 10^{-5}$ for average response, bootstrap analysis, 'Materials and methods'). Significant responses were also observed at 54% of the multiunit recording sites (n = 7/13), and in 30% of AIV neurons that did not meet the criteria for antidromic identification (n = 6/20 neurons). The response of AIV neurons to noise bursts was brief, occurring with an average latency of 23 ± 12 ms from noise onset and having an average duration of 90 ± 43 ms ('Materials and methods', *Figure 7L*). There was no significant difference between the responses of antidromically identified neurons and those that were not antidromically identified. Thus, these data are pooled in the statistics on latency and duration.

Are AIV neurons responsive to noise bursts only during singing, or can similar auditory responses be evoked during non-singing? We recorded the responses of AIV neurons to noise bursts presented during playback of the bird's own song (BOS) in 25 identified VTA/SNc-projecting AIV neurons under anesthesia (n = 3 birds), and another 28 such neurons in freely behaving birds (n = 4 birds). Under these conditions, noise bursts elicited no significant response in any of these neurons (*Figure 8A,C,E*, 'Materials and methods'). We also examined whether AIV neurons respond to noise bursts presented in a silent background (non-singing, no BOS). In this case, auditory responses were observed in a small fraction of neurons in both awake and anesthetized birds: Altogether, 12/53 neurons exhibited a significant response within a 1 s window after noise onset (bootstrap, $p < 0.02$, 'Materials and methods'). However, these responses were significantly slower and exhibited longer latency than during singing ($p < 0.001$, *Figure 8B,D,E*, only one neuron responded significantly within 150 ms of noise onset). In comparison, all the neurons that were found to be noise-responsive during singing (significant activity within a 1 s window), exhibited this response within 150 ms of the noise burst onset. Altogether, we have found that the auditory responsiveness of AIV neurons to disruptive auditory stimuli is highly state-dependent, exhibiting fast and robust firing-rate changes only during singing.

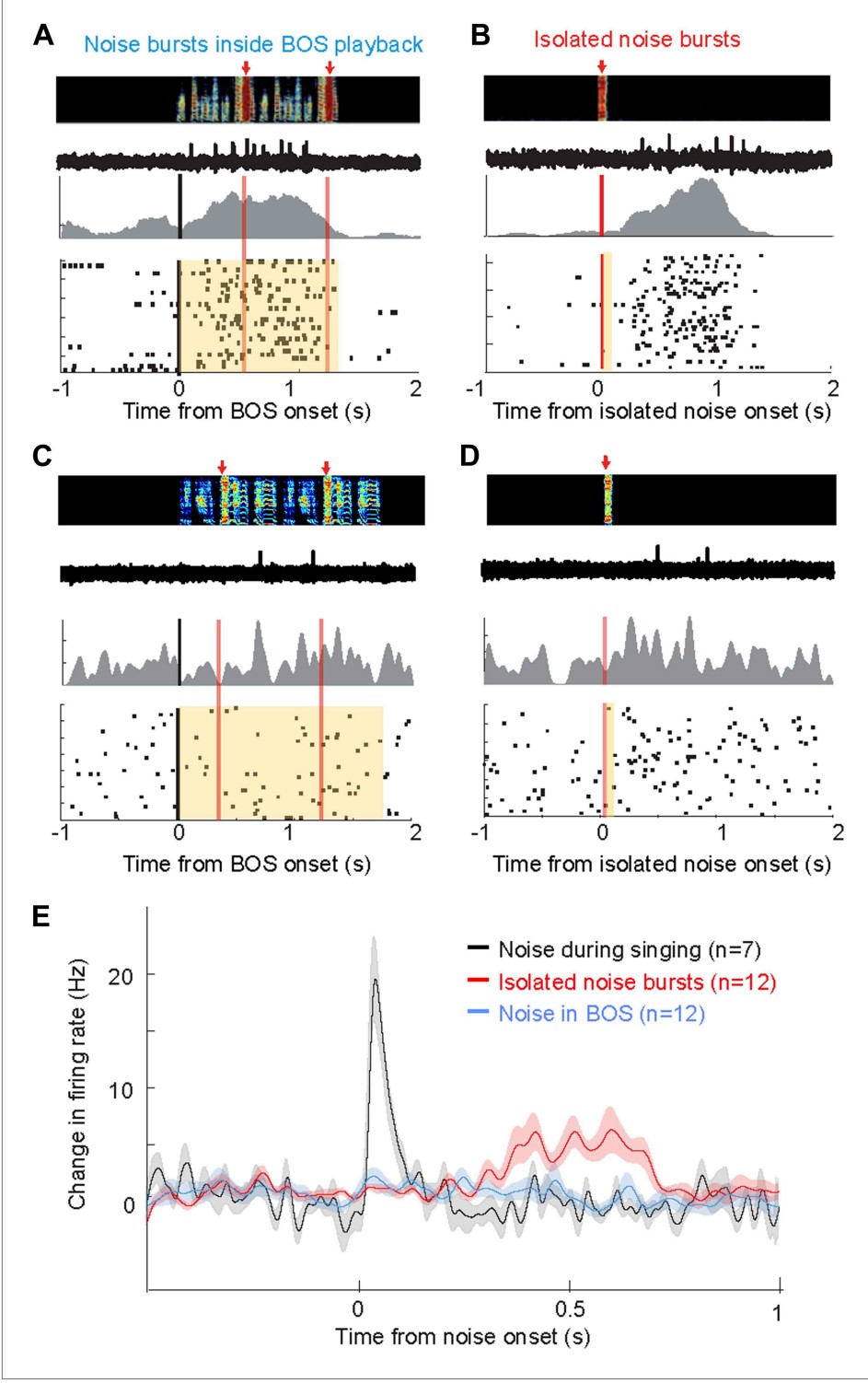

**Figure 8**. Response of AIV neurons to noise bursts during non-singing. (**A**) Activity of an antidromically identified AIV neuron recorded during presentation of noise bursts during playback of birds own song (BOS). Top to bottom panels: song spectrogram and simultaneous recording of neuronal activity; raster plot and spike histogram aligned to BOS onset (black line). Time of noise bursts indicated by vertical red lines. Yellow band denotes period of BOS playback. Note lack of response to noise bursts. (**B**) Response of the same neuron in panel **A** to presentation of isolated noise bursts (no BOS, no singing). Note slow timecourse of the response. (**C** and **D**) Recording of another antidromically identified AIV neuron (notation same as in panels **A** and **B**). (**E**) Averaged PSTH for all VTA/SNc-projecting AIV neurons
*Figure 8. Continued on next page*

*Figure 8. Continued*

that exhibited a significant response to isolated noise bursts within 1 s after noise onset. Average response of these same neurons to noise bursts presented during BOS playback (blue). In comparison, averaged response is shown for all VTA/SNc-projecting AIV neurons that responded significantly to noise burst during singing (black trace).

## Discussion

Motivated by the hypothesized role of dopaminergic signaling in reinforcement learning (*Houk et al., 1994*; *Schultz et al., 1997*; *Bayer and Glimcher, 2005*; *Tsai et al., 2009*), we set out to examine the role in vocal learning of a recently discovered songbird cortical area (*Gale et al., 2008*) that projects to VTA and SNc. We have characterized the spatial extent of neurons within the intermediate arcopallium that project to VTA and SNc, and refer to the region of arcopallium retrogradely labeled from these midbrain dopaminergic areas as AIV. Using a combination of anatomical and electrophysiological techniques, we have elucidated the afferent inputs to AIV neurons from other cortical areas. We examined the effect on vocal learning of lesions targeted to AIV, and carried out electrophysiological recordings of AIV neurons. Our findings are broadly consistent with the hypothesis that an arcopallial region with descending auditory cortical projections, both to the dopaminergic midbrain and to midbrain and brainstem auditory centers, plays a role in vocal learning.

The area we have identified as AIV is adjacent to other structures that have been hypothesized to play a role in vocal learning (*Bottjer and Altenau, 2010*). The dorsal arcopallium (also referred to as the lateral intermediate arcopallium, LAI, by *Jarvis et al., 2013*), receives inputs from the nidopallium adjacent to LMAN (LMAN-shell), and projects to the basal ganglia, motor thalamus, tectum, and the reticular formation (*Bottjer et al., 2000*). Our anatomical findings suggest that AIV, while adjacent to Ad, is part of a distinct, largely non-overlapping, anatomical circuit. In contrast to Ad, AIV does not receive a projection from LMAN-shell; rather it receives inputs from auditory cortical areas (CM, L1, and HVC-shelf) and caudal nidopallium. Furthermore, our findings show that AIV, but not Ad, projects to VTA/SNc. Given its anatomical overlap with RAcup, parts of AIV also likely project to MLD or to the shell of the thalamic nucleus ovoidalis (*Mello et al., 1998*).

Earlier studies have presented conflicting views on the function of Ad/LAI, particularly in regard to its role in vocal learning (*Bottjer et al., 2000*; *Achiro and Bottjer, 2013*). *Bottjer and Altenau (2010)* describe evidence that lesions of Ad produce deficits in vocal learning, not unlike those we report here for lesions of AIV. In contrast, Feenders et al. showed immediate early gene activation in Ad/LAI and in LMAN-shell (which projects to Ad/LAI) during locomotory activity (i.e., hopping), but find no evidence for activation in these areas during singing. This led them to suggest that Ad/LAI and RA form the output of a general avian cortical motor circuit, of which RA is a component specialized for vocal production (*Feenders et al., 2008*).

Our findings are consistent with the view that AIV, but not Ad, play a role in vocal motor learning. We found that lesions of AIV caused deficits in vocal imitation, while lesions of Ad did not cause any such deficit, suggesting that the loss of song imitation following lesions targeted to AIV was not a secondary consequence of an unintended lesion in Ad. Our finding that larger lesions of Ad produced severe akinesia and immobility is consistent with the hypothesis that Ad plays a role in locomotion and other motor behaviors, rather than vocal learning (*Feenders et al., 2008*).

There are several potential explanations for the discrepancy between our findings on the effects of Ad lesions and those of *Bottjer and Altenau (2010)*. Clearly, the Ad lesion protocol of Bottjer and Altenau impacted a subset of neurons important for vocal learning, while our Ad lesion protocol did not. This subset of neurons, in principle, could be within Ad, within AIV, or could be in another, as yet undescribed, adjacent region important for vocal learning. In our view, the most parsimonious explanation for the detrimental effects on vocal learning is that the Ad lesions of Bottjer and Altenau had an unintended impact on neurons in AIV. More detailed electrophysiological studies of the pathways into and out of Ad will be needed to completely resolve these different views of its function.

The effects of lesions directed to AIV in juvenile birds bear some resemblance to the effects of lesions in the AFP, a basal ganglia–thalamocortical circuit necessary for vocal learning. Lesions of Area X, the basal ganglia component of the AFP, in young birds have a substantial detrimental effect on song imitation and result in persistent song variability into adulthood (*Scharff and Nottebohm, 1991*).

In contrast, lesions of Area X have relatively little immediate effect on song performance in adult and juvenile birds (*Goldberg and Fee, 2011*; *Kojima et al., 2013*). Similarly, we found that birds in which AIV was lesioned at an early age produced a poor imitation of the tutor song and developed a less stereotyped song than did control birds, and AIV lesions in adult or juvenile birds have little immediate effect on vocal performance. Lesions of Area X also largely block the degradation of song following deafening (*Kojima et al., 2013*), and similarly, but to a lesser extent, we found that lesions of AIV slow song degradation in deafened birds. The similar patterns of findings in AIV-lesioned birds and Area X-lesioned birds is consistent with the possibility that the AIV, or some components of it, may interact with the AFP, perhaps by transmitting to it a signal important for its function.

In principle, there are several ways the loss of a descending auditory cortical projections might affect vocal learning. We first address the possible role of the projection from AIV to the dopaminergic midbrain in vocal learning. In the songbird, dopaminergic inputs to Area X have been hypothesized to carry signals related to motivational aspects of singing and to regulate vocal variability, in particular while switching between less variable song directed to a female bird to more variable song produced in social isolation (*Yanagihara and Hessler, 2006*; *Hara et al., 2007*; *Leblois et al., 2010*). Notably, we find that juvenile and young adult birds exhibit normal song variability after AIV lesions, suggesting that AIV may not be involved in the regulation of vocal variability, and that the deficits in vocal learning do not result from a loss of exploratory song variability. Alternatively, reduced glutamatergic drive after AIV lesion could result in reduced tonic dopaminergic input to Area X. In mammals, reduced dopaminergic input to the striatum can have deleterious effects on BG function, including reduced spontaneous activity and the loss of spines in the medium spiny neurons (*Kreitzer and Malenka, 2008*). Songbirds exhibit patterns of dopamine receptor expression in cortex and Area X (*Kubikova et al., 2010*), and responses to dopamine stimulation (*Ding and Perkel, 2002*) that parallel those found in mammals. Finally, another possibility is that impoverished vocal imitation after lesions of AIV results from the loss of a reinforcement signal that carries information about recent song performance. The short-latency response of AIV neurons to disrupted auditory feedback is consistent with the possibility that AIV may play a role in computing or transmitting a fast online signal to VTA/SNc.

Previous studies have shown that neurons in the vicinity of RA, and potentially within AIV, have several sub-telencephalic targets other than VTA/SNc, including a higher-order auditory thalamic nucleus Ov-shell and parts of the auditory midbrain (*Vates et al., 1996*). Thus, deficits in vocal learning following lesions targeted to AIV could potentially arise from loss of signaling in these other pathways. Further studies will be required to elucidate the relation between neurons projecting to these different downstream targets, and to determine the role of these different pathways in vocal learning.

One key to understanding the function of AIV is to characterize its cortical afferents, and several candidates have been identified. Güntürkün et al. have described a projection in pigeons from the lateral caudal nidopallium (NCL) to the ventral arcopallium (*Kroner and Gunturkun, 1999*). The projection we describe here in the zebra finch—from the caudal nidopallium (NC) posterior to HVC—terminates primarily in the posterior 'stem' part of AIV. This projection may be related to the previously described nidopallium caudolaterale (NCL) projection, although NCL appears to be more lateral than NC. NCL was recently found to be involved in error and reward processing (*Starosta et al., 2013*), and based on several lines of evidence, it has been suggested that NCL may be analogous to mammalian prefrontal cortex (*Kroner and Gunturkun, 1999*). Notably, in mammals the major cortical projection to VTA arises from prefrontal cortex, a key structure for cognitive control and goal-directed actions (*Murase et al., 1993*).

Previous studies in the zebra finch have also identified inputs to the intermediate arcopallium from auditory cortical regions, including primary auditory cortical field L1 and HVC-shelf in the nidopallium (*Kelley and Nottebohm, 1979*; *Vates et al., 1996*; *Mello et al., 1998*). These projections were found to terminate near RA, in a region termed RA-cup, and are likely homologous to projections in the pigeon from the dorsal nidopallium (Nd) to the ventromedial intermediate arcopallium (AIvm) (*Wild et al., 1993*). We have confirmed the existence of these projections, and using anatomical tracing and electrophysiological techniques, we have demonstrated that these inputs directly or indirectly innervate AIV neurons projecting to the dopaminergic midbrain. Inputs from L1 and anterior parts of HVC shelf appear localized to the parts of AIV anteroventral to RA. Inputs from posterior HVC shelf appear to terminate in both the anterior and posterior parts of AIV, and show significant overlap with inputs from the more posterior caudal nidopallium described above.

Our anatomical studies have also revealed a novel projection to AIV from the caudal mesopallium (CM), an avian auditory area previously identified as homologous to higher-order auditory cortex

(*Bauer et al., 2008*) or to upper layers of primary auditory cortex (*Karten, 2013*). The projection from CM terminates within AIV, and electrical stimulation in CM causes spiking activity in VTA/SNc-projecting AIV neurons, likely through orthodromic activation. The existence of a projection to AIV from CM and L1 may be important for two reasons. The cluster of neurons in CM retrogradely labeled from AIV was overlapped with the region of CM, also known as Avalanche (Av), that forms bidirectional synaptic interactions with both HVC and NIf (*Bauer et al., 2008*; *Akutagawa and Konishi, 2010*), two song control nuclei also involved in vocal learning (*Roberts et al., 2012*). Furthermore, a small subpopulation of neurons in CM and L1 have been shown to respond to noise bursts presented during singing, but not to noise bursts presented during playback of the birds own song (*Keller and Hahnloser, 2009*). These responses, highly reminiscent of responses in AIV, have led to the suggestion that one computational function of CM and L1 may be to compare actual auditory feedback with the song template (*Keller and Hahnloser, 2009*). Further studies are required to determine if the hypothesized error-related responses of AIV neurons derive from activity in this population of CM and L1 neurons.

One can speculate about the potential role for AIV in computing or transmitting a fast online signal to VTA/SNc that potentially carries information about recent song performance. Most VTA/SNc-projecting AIV neurons responded to disruptive auditory feedback with a latency of less than 25 ms and response duration of less than 100 ms. This response may have a sufficiently fast temporal resolution to mediate vocal learning, assuming that the synaptic learning rules in Area X and the motor pathway employ a short-term synaptic memory such as an eligibility trace (*Sutton and Barto, 1998*; *Fee and Goldberg, 2011*; *Redondo and Morris, 2011*). Such a fast dopaminergic reinforcement signal could, in principle, be used in Area X to correlate vocal performance with an efference copy of vocal motor commands (*Fee and Goldberg, 2011*) and to drive corticostriatal plasticity at HVC inputs to medium spiny neurons (MSNs). The MSNs could then act through their downstream pallidothalamic pathway to (1) bias the motor system in favor of vocal commands that previously led to better song performance and (2) drive plasticity in the motor pathway such that the juvenile song gradually converges to the desired vocal output (*Andalman and Fee, 2009*; *Fee and Goldberg, 2011*; *Warren et al., 2011*; *Charlesworth et al., 2012*).

Additional studies would be required to establish the nature of auditory reinforcement signals in the pathway we have described here, or possibly through other downstream targets, and to establish a causal role for these circuits in vocal learning.

## Materials and methods

### Subjects

Animal subjects were male zebra finches (n = 110) (45–120 days post hatch, dph). Birds were obtained from the Massachusetts Institute of Technology zebra finch breeding facility (Cambridge, Massachusetts). The care and experimental manipulation of the animals were carried out in accordance with guidelines of the National Institutes of Health and were reviewed and approved by the Massachusetts Institute of Technology Committee on Animal Care.

### Identification of AIV

Retrograde labeling of neurons in AIV was obtained by injecting fluorescently labeled cholera toxin β subunit (Molecular Probes) into the ventral tegmental area (VTA) and substantia nigra pars compacta (SNc). VTA/SNc were localized stereotaxically with reference to the principal thalamic auditory relay nucleus Ovoidalis, as described below.

### Reliable targeting of VTA/SNc

We have developed a technique that allows us to target VTA and SNc with high reliability, based on the mapped location of the thalamic auditory relay nucleus Ovoidalis (Ov). Specifically, the head was oriented with the flat anterior portion of the skull rotated forward at a pitch of 50°. Using an extracellular electrode (Carbostar-1, Kation Scientific, Minneapolis, MN), tilted in the LM direction 2° towards the midline, Ov was located and mapped based on its robust auditory responses. Injections were made into VTA relative to Ov as follows: 300 µm anterior from the center of Ov, 200 µm medial from the medial edge of Ov, and 1800 µm ventral to the middle of Ov in the DV direction. Injections into SNc were targeted 350 µm posterior and 350 µm lateral to the VTA coordinates.

For electrophysiology experiments requiring antidromic stimulation in VTA/SNc, bipolar stimulating electrodes were placed such that, to the extent possible, one wire was in VTA and the other wire was in SNc.

## Excitotoxic lesions of AIV

Bilateral lesions of AIV were carried out by injecting 2% N-Methyl-DL aspartic acid (NMA, Sigma, St Louis, MO) at multiple injection sites in ventral arcopallium around RA chosen to maximally lesion VTA/SNc-projecting neurons. All AIV lesions were carried out stereotaxically. The head was oriented with the flat anterior portion of the skull rotated forward at a pitch of 70°. Because a large portion of AIV is located ventral to RA, access to AIV was achieved by a laterally rotated penetration. The head was rotated 45° (roll) to the left for lesions of AIV in the right hemisphere, and rotated to the right for lesions of AIV in the left hemisphere. Using a carbon fiber electrode (Mod #E1011-20; Carbostar-1, Kation Scientific.) RA was then completely mapped based on its characteristic electrophysiological signature (regular spiking interspersed with bursts). Once all of the edges of RA were localized in the rotated frame, these were used to calculate the coordinates of the injection sites. In the injection coordinates given below (ML, AP, DV), dimensions are in μm; positive numbers represent more lateral, more anterior, and more ventral displacements, respectively. The coordinates are specified relative to the lateral edge of RA in the medial–lateral direction, relative to the center of RA in the anterior–posterior direction, and relative to the most ventral edge of RA in the dorsal–ventral direction. Injections were made through a glass pipette using a digitally controlled injection system (Nanoject, Drummond Scientific, Broomall, PA). Injections at each site were made in multiple boluses of 13.8 nl, which were spaced at 5-min intervals to avoid backflow along the pipette. AIV lesions (n = 17 birds) were created by up to seven different injection sites (in five different penetrations) at the coordinates specified below. The last number in each coordinate indicates the number of injection boluses made at that site: (+200,−400,+200; 4); (+300,+0,+200; 4); (+200,+300,+0; 4); (+0,+700,+0; 2); (+0,+700,+300; 2); (−300,+900,+0; 2) ; (−300,+900,+300; 2).

Note that, given the complex spatial structure of AIV, it was difficult to achieve a complete lesion of VTA/SNc projecting neurons. For example, we did not attempt to lesion the thin shell of AIV neurons dorsal to RA. Furthermore, our lesions may have affected areas of the intermediate arcopallium adjacent to AIV.

## Excitotoxic lesions of Ad

We targeted the portion of Ad most likely affected by our AIV lesion procedure (n = 7 birds) as follows: Bilateral lesions of Ad were carried out stereotaxically by injecting 1% NMA at multiple injection sites. The stereotaxic procedure was similar to that described above for AIV lesions. RA was mapped electrophysiologically to determine the lateral border and center (in the AP direction) of RA. Ad was also mapped electophysiologically by its characteristic bursting patterns, which distinguished it from the overlying arcopallium. Injection sites are listed below: the (ML, AP) coordinates are given in the same notation given above for the AIV lesion. However, the injections were targeted to the center of Ad in the dorso–ventral dimension as determined from electrophysiological mapping of Ad. Injections at each site were made in 3 boluses of 13.8 nl, which were spaced at 5-min intervals to avoid backflow along the pipette. Injections were made at the following locations: (+250,−100); (+250,+300); (+650,−100); (+650,+300); (+1000,−100); (+1000,+300). This lesion protocol typically spared the lateral-most ~200 μm of Ad (see *Figure 5A*).

To determine the effects of complete lesions of Ad on vocal learning, we also carried out (n = 3 birds) an Ad lesion protocol with two additional lateral injection sites. Injections were made at the following locations: (+250,−100); (+250,+300); (+650,−100); (+650,+300); (+950,−100); (+950,+300); (+1200,−100); (+1200,+300). These birds exhibited severe immobility after surgery, and failed to eat or drink, requiring early termination of the experiment.

## Developmental study of song imitation (*Figure 4*)

Male juvenile birds were maintained in the aviary in their home cages with both parents until the age 44–45 dph, at which point they were transferred to sound isolation/recording chambers until they began to sing and baseline song was acquired. For experimental birds, bilateral AIV or Ad lesions were carried out as described above. After surgery, birds typically began to sing within 5 days (range 2–5 days). Birds were maintained in isolation and recorded continuously until they reached the age of 90 dph. Once it was confirmed that post-90 dph, songs had been recorded, we histologically verified the

extent of AIV lesions, as described below. Unmanipulated control birds were isolated at 44–45 dph (as for experimental birds) and were maintained in sound isolation/recording cages until they reached the age of 90 dph, or until post-90 dph song had been acquired. Assignment of birds to unmanipulated control or experimental groups was initially made randomly, but subsequent juveniles from a given breeding cage (with a given tutor) were placed in control or experimental groups to provide sibling matches between these conditions.

## Histological verification of AIV lesions

AIV lesions were examined histologically at the end of each experiment. Retrograde tracer (fluorescently labeled cholera toxin β subunit, Molecular Probes) was injected into VTA/SNc and 4 days later the bird was sacrificed and perfused transcardially with saline and then 4% paraformaldehyde. The brain was removed and post-fixed in 4% paraformaldehyde overnight. The brain was sectioned, stained for NeuN, and imaged under a fluorescence microscope (Zeiss Axioplan, Germany). The extent of AIV lesions was analyzed on the basis of loss of NeuN staining and the density of retrogradely labeled neurons in these regions. Quantification was based on the size of the lesion in each parasagittal section extending from the medial edge of RA to 200 µm past the lateral edge, and yielded two numbers indicating the fraction of AIV lesioned on the left and right side. The extent of any damage to RA was also monitored and birds with more than 5% RA lesion were excluded from the analysis (n = 2). The histological assessment of lesion size was conducted blind to the behavioral effects on song, and the quantification of the behavioral effects was done blind to the histological analysis of lesion size.

## Testing the functionality of anatomical projections

To test the functional connectivity of projections from L1, CM, and HVC-shelf to VTA/SNc-projecting neurons in AIV, we implanted bipolar stimulating electrodes in each of these auditory cortical areas (L1, n = 2 birds; CM, n = 3 birds; HVC-shelf, n = 3 birds). A bipolar stimulating electrode was also placed in VTA/SNc to antidromically identify neurons in AIV. Guided by the results of the anterograde tracing experiments (*Figure 2*, *Figure 2—figure supplement 1*), recordings were made in the regions of AIV ventral and anterior to RA using a Carbostar electrode (1 MOhm, Kation Scientific). Within this region, single-unit and multi-unit activity was broadly responsive to stimulation in both VTA/SNc and in L1, CM, or HVC-shelf. Threshold currents required to elicit spiking responses were between 50 and 200 µA (single 0.2-ms unipolar pulses). Location of the stimulating electrodes was verified histologically.

## Analysis of song imitation

We modified the previously published Song Analysis Pro (SAP) algorithm (*Tchernichovski et al., 2000*) and developed an automated procedure for comparison of pupil songs with the tutor motif (Software is available at *Mandelblat-Cerf and Fee, 2014*). Following SAP, songs were represented by acoustic features. Similarity between time points in the two songs is based on the Euclidean distance in the feature space between these points. The procedure builds a similarity matrix for all possible pairs of time points between the two song samples.

To quantify tutor–pupil similarity the tutor motif was segmented into separate syllables. While syllable segmentation is highly reliable in tutor song, it is less reliable in the more variable pupil song, which was thus segmented into equally sized sections, each twice the duration of the tutor motif. For each tutor syllable our procedure uses the similarity matrix described above to find the sections of the pupil song bout that have the highest similarity and assigns an overall similarity score. For each match between a tutor's syllable and a pupil's song section, a sequencing score is evaluated by computing the match of the next syllable in the tutor song with the next section of the pupil song. The algorithm then computes a composite 'tutor imitation score' as the product of song similarity and sequence similarity.

For each bird we examined song spectrograms and computed the imitation scores for the first day of singing after isolation, and before any further manipulation. Three birds exhibited clear evidence of tutor imitation and were excluded from further experiments.

## Imitation capacity

To determine the fractional extent to which AIV-lesions reduced the capacity of birds to imitate their tutor song, imitation scores of each bird were placed on a linear scale ranging from the average imitation score of control birds 0.197 ± 0.011 to the average similarity score from all pairwise comparisons

of 20 unrelated adult birds in our colony 0.104 ± 0.0027, yielding a total dynamic range of 0.093 [=0.197–0.104] ± 0.00636. For birds in which we estimate that more than half of AIV was lesioned, the average imitation score was 0.116 ± 0.008. Therefore, we can compute the similarity of AIV-lesioned birds to their tutor songs relative to the song similarity of unrelated birds, as a fraction of the total dynamic range, as 12.9% [=(0.116–0.104)/0.093] ± 9.2%. Thus, AIV-lesioned birds suffered a loss of imitation capacity compared to controls of 87% [=1.0–0.129] ± 9%.

## Quantification of song degradation following deafening and/or AIV lesion (*Figure 6*)

Male juvenile birds were obtained from the aviary at the age 65–70 dph and were maintained in sound isolation cages until they began to sing. Upon reaching age the 75–80 dph, pre-surgery songs were recorded, and birds were randomly placed in three experimental groups: deafening, AIV-lesion, and combined deafening and AIV-lesion. Deafening was achieved by bilateral cochlear extirpation, which was carried out by accessing the cochlea through the posterio-lateral cranium. AIV lesions were carried out as described above. Upon recovery from surgery, birds were maintained in sound isolation cages. Undirected song was recorded continuously for 2 weeks.

To quantify song degradation, song was recorded every day post-surgery and compared to song recorded prior to surgery. Self-similarity scores were computed using the same algorithm used to evaluate tutor imitation.

## Chronic neural recordings in AIV

Single-unit recordings of VTA/SNc-projecting AIV neurons were obtained from seven young adult birds (90–120 dph) during undirected singing. For antidromic identification of AIV neurons, bipolar stimulating electrodes were placed in VTA/SNc, according to the methodology described above. AIV was localized with reference to the boundaries of RA, as electrophysiologically identified by the characteristic tonic activity of RA neurons. Placement of the chronically implanted electrode array was further refined by finding the regions in AIV that exhibited strong antidromic responses to brief unipolar square current pulses (0.2 ms duration) applied to the stimulating electrodes in VTA/SNc. Peak currents up to 350 µA were used to search for antidromically activated neurons: threshold currents for antidromic activation of neurons in AIV ranged from 90 µA to 250 µA. Recordings in AIV were made using an array of three recording electrodes (200 µm spacing, Microprobe, Gaithersburg, MD, #PI20033.0A3, 3 MΩ), mounted in a motorized microdrive (n = 20 birds implanted). Data were acquired and analyzed using custom Matlab software.

Disruptive auditory stimuli consisted of 50-ms duration pulses of white noise (95 dB$_{SPL}$) similar to that previously used to drive song plasticity (*Tumer and Brainard, 2007*; *Andalman and Fee, 2009*), which were presented with random timing during undirected singing.

## Recordings during playback of birds own song (BOS)

### Auditory stimuli

A week before surgery, birds were isolated and undirected song was recorded. The song motif was extracted and a BOS stimulus was constructed from concatenation of 2 motifs. To mimic the auditory input during disruptive auditory feedback during singing, we constructed 'BOS with noise' stimulus in which noise bursts were added to a single syllable within each repetition of the motif. For each bird, the noise burst was added at a fixed time near the middle of the motif. The peak loudness of the BOS playback was set to be equal to the average peak loudness of song measured on the head of the singing birds (90 dB$_{SPL}$) (*Andalman and Fee, 2009*). Noise burst loudness was the same as for the presentation of disruptive auditory feedback during singing (95 dB$_{SPL}$).

### Recordings in anesthetized birds

Birds were anesthetized with three intramuscular injections of 25 µl each of 20% urethane administered at 20 min intervals. If needed, another injection was given during the recording session. Playback of bird's own song (BOS) was delivered to the birds using miniature custom-built headphones (E-A-RTONE Mod-3A Audiometric Insert Earphone, Indianapolis, IN). Sound was played at 70dBSPL and 90dBSPL, calibrated using a probe microphone (Model ER-7C, Etymotic, Inc, Elk Grove Village, IL). VTA/SNc-projecting AIV neurons were antidromically identified using the protocol described above for chronic recordings.

## Recordings in awake (non-singing) birds

A motorized microdrive was implanted for recordings in AIV as described above. After the surgery, the bird was maintained in a sound attenuation chamber. VTA/SNc-projecting AIV neurons were recorded during playback of BOS and BOS with noise on alternating presentations. Sound level for BOS playback was 90 dBSPL and calibrated using a Bruel & Kajer Model 2236 sound level meter placed at approximately the center of the bird cage. Histological verification of electrode positions was carried out after the recordings. Data were collected from four birds.

## Analysis of neuronal responses

The significance of single neuron and multiunit responses to noise bursts were analyzed in two different ways: the first was a comparison of the spike counts in a window 150 ms before vs 150 ms after noise onset (paired *t* test). The second was a test for a significant peak in the peri-stimulus time histogram (PSTH) using a bootstrap analysis: over a 2 s window (1 s before stimulus onset to 1 s after stimulus onset) spikes were shifted circularly, where the extent of the shifts were drawn from a uniform distribution of ±1 s. Given the resulting PSTH, the peak within 1 s after stimulus was noted. This random shuffling was repeated 1000 times to obtain the distribution of peaks in the shuffled data sets. Neurons that exhibited a real PSTH peak that exceeded the 95th percentile of the shuffled data were considered significant. The significance of song-locked firing rate modulations of AIV neurons during singing was analyzed similarly. The highest observed peak in the motif-aligned PSTH was compared to the distribution of PSTH peaks obtained from surrogate data sets in which spike times were circularly within the motif.

## The significance of the averaged response to noise bursts of all projecting neurons

Over a 2 s window (1 s before noise onset to 1 s after noise onset) the averaged response of each neuron was shifted circularly, where the extent of the shifts were drawn from a uniform distribution of ±1 s. Given the resulting averaged response across all neurons, two values were noted: the peak in the firing rate and the maximal spike count over 150 ms. This random shuffling was repeated 1000 times to obtain the distributions in the shuffled data sets. The peak of the measured averaged response within 150 ms from noise onset, and the spike count over this period, were tested against the distribution of peaks and spike counts. The indicated p-value is the higher of these two tests.

## Response latency and duration

Activity around stimulus onset was binned in 2 ms bins. Each bin after stimulus onset was tested against the distribution of the activity prior to the stimulus, using a z-test. Response onset (latency) was detected by the first bin for which the following five consecutive bins (10 ms) were significantly different than the activity prior to the stimulus (z-test, all with p<0.05). To assess response duration, we searched for the first bin after response onset for which, the following 10 consecutive bins (20 ms) were insignificantly different than the pre-stimulus activity (z-test, all with p>0.05).

## Stereotypy of identified syllables

To evaluate syllable stereotypy, independent of sequence stereotypy, we used the peak of the spectral cross-correlation (*Aronov et al., 2008*), on a syllable-by-syllable basis. A database of song syllables was constructed for each bird (n = 500–600 syllables). The spectrogram of each syllable was computed using the multi-taper method (k = 2 tapers, bandwidth parameter p=1.5, 10 ms window, 1 ms step size). 100 syllables were randomly selected (without replacement) from the data set. For each selected syllable, a sample of 100 syllables was drawn randomly from the data set. The similarity of the selected syllable to each of the sampled syllables was evaluated following the spectral correlation method: first, the spectrogram correlation matrix was computed from the correlation of power spectra (860 Hz and 8.6 kHz) between each 1 ms time slice in the selected syllable and each 1 ms time slice in the sampled syllable. Second, a lag correlation function was computed as the sum along the diagonals of the correlation matrix. Third, the maximum of the lag correlation function was determined. The values resulting from all comparisons to the selected syllable formed a distribution. The value of the 95 percentile of this distribution represents the stereotypy of the selected syllables. For each bird, the syllable stereotypy metric (*Figure 5D*) was computed as the average over all 100 selected syllables.

## Acknowledgements

We thank Jesse Goldberg for helpful suggestions. We also thank Daniel Rubin, Anton Konovchenko, and Tim Currier for technical assistance. Funding for this work was provided by the National Institutes of Health (R01 MH067105) and McGovern Institute internal funding.

## Additional information

### Funding

| Funder | Grant reference number | Author |
| --- | --- | --- |
| National Institutes of Health | R01 MH067105 | Michale S Fee |

The funder had no role in study design, data collection and interpretation, or the decision to submit the work for publication.

### Author contributions

YM-C, LL, ND, MSF, Conception and design, Acquisition of data, Analysis and interpretation of data, Drafting or revising the article, Contributed unpublished essential data or reagents

### Author ORCIDs

Yael Mandelblat-Cerf, http://orcid.org/0000-0002-6356-5856

### Ethics

Animal experimentation: This study was performed in strict accordance with the recommendations in the Guide for the Care and Use of Laboratory Animals of the National Institutes of Health, were reviewed and approved by the Massachusetts Institute of Technology Committee on Animal Care (IACUC). Permit Number: 0712-071-15. Every effort was made to minimize suffering.

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
