## [Decision Letter]

[Editors’ note: although it is not typical of the review process at *eLife*, in this case the editors decided to include the reviews in their entirety for the authors’ consideration as they prepared their revised submission.]

Thank you for sending your work entitled “A descending auditory cortical projection is necessary for vocal learning in a songbird” for consideration at *eLife.* Your article has been favorably evaluated by a Senior editor and 3 reviewers, one of whom is a member of our Board of Reviewing Editors.

The following individuals responsible for the peer review of your submission have agreed to reveal their identity: Ronald L. Calabrese (BRE) and Marc Schmidt (reviewer).

The Reviewing editor and the other reviewers discussed their comments before we reached this decision, and the Reviewing editor has assembled the following comments to help you prepare a revised submission. The full initial reviews are appended below as well, and many of the minor comments found in them should be addressed.

Overview:

In this manuscript the authors present careful anatomical, electrophysiological, and behavioral analyses in songbirds of a brain area they term ventral intermediate arcopallium (AIV), which receives auditory cortical input and has output to the dopaminergic VTA/SNc. They further show that AIV neurons that are demonstrated to project to VTA by antidromic activation respond to disruptive auditory feedback during singing but not during the playback of the bird's own song or the bird's own song with disruptive auditory feedback by in vivo single unit (and multiunit) recording in awake unrestrained birds during singing and quiescence. They also show that lesions of the AIV leads to deficits in vocal learning but do not alter song in mature birds. The data set presented is extensive and the analyses quantitative. The AIV neurons described provide an opening to the dopaminergic reward system that promises in future to help elucidate how auditory feedback can alter vocal output: a significant problem in song birds and humans. This analysis will generate wide interest in the songbird and sensory motor integration communities.

Concerns:

There are several concerns in the reviews by the BRE and the external reviewers that are provided to the authors to help them in revision. To focus this revision and remove any apparent conflict in the reviews, a consensus has formed about three areas.

Anatomy:

The reviewers are aware that as work has proceeded in the songbird, focus has shifted from distinct nuclei like RA or X to more diffusely organized areas like the area described here as AIV. AIV must be as precisely delimited as is possible and the authors should be at pains to do so. The anatomy presented in support of AIV must be discussed in the context of past findings, particularly [8] (Parallel pathways for vocal learning in basal ganglia of songbirds. Nature Neuroscience 13, 153-155.), even if this makes sections of the Discussions more for birdsong experts. It would also serve the community if some other nearby areas could be differentiated from or included in AIV e.g. RA cup. This is not to set previous work as some anatomical standard but rather to motivate further clarification that will be healthy for the field. Not only should the authors specify exactly what the boundaries of AIV are but they should thoroughly discuss how it is related to Ad. They need to state specifically why they think they have lesioned one and not the other and how the AIV vs. Ad lesions lead to different or similar behavioral results. They should try to explain discrepancies in their data and that from [8].

Many of the projections described to AIV have been previously described, at least in part. The projection from CM to AIV is novel and potentially most functionally significant for the physiological responses of AIV to VTA projecting neurons; it should be highlighted.

Behaviour:

The authors should substantially increase the presentation of their primary data on song that is encompassed in their imitation score. There are no figure limitations in *eLife*, so it is possible to present more sonograms so that other workers can more easily judge how song is affected or not affected by a lesion. In addition to changes in imitation score, they should give a qualitative description of how song is affected by their lesions. They should then thoroughly discuss how the results are similar to or different than Ad and X lesions by others.

Physiology:

There are a number of specific comments by Reviewer #2 that should be addressed.

The Abstract is written quite conservatively and is highly appropriate; it might be best to adjust the title so that it more matches the conservative tone of Abstract.

Other issues to address:

Reviewer #1:

In this manuscript the authors present careful anatomical, electrophysiological, and behavioral analyses in songbirds of a previously unstudied brain area (ventral intermediate arcopallium (AIV)). They show that AIV receives auditory cortical input and has output to the dopaminergic VTA/SNc. They further show that AIV neurons respond to disruptive auditory feedback during singing but not during the playback of the bird's own song or the bird's own song with disruptive auditory feedback by in vivo single unit (and multiunit) recording in awake unrestrained birds during singing and quiescence. They also show that lesions of the AIV leads to deficits in vocal learning but do not alter song in mature birds. The data set presented is extensive and the analyses thorough and quantitative.

This AIV neurons discovered provide an opening to the dopaminergic reward system that promises in future to help elucidate how auditory feedback can alter vocal output: a significant problem in song birds and humans. This elegant analysis will generate wide interest in the songbird and sensory motor integration communities.

Concerns:

1) The work is carefully done with detailed analysis and the figures appropriately illustrate the data. Nevertheless the figures are very complicated and could benefit from more detailed legends and perhaps better labeling. A tremendous amount of data is presented in supplementary figures. *eLife* discourages use of supplementary data and rather includes all important data in the text. The authors should ruthlessly select the relevant data from the supplementary data and present those integrated into the text. For example the lower panels in Figure 5—figure supplement 5 could be eliminated. I defer to the expert reviewers and the authors on what is essential, however.

2) A better general explanation (and figure) of the system than afforded in the Introduction, Discussion and by Figure 1 would greatly help the general reader especially in Discussion. I fully understand why Figure 1 is simplified to explain the anatomical experiments, but by the time we move beyond these experiments through Results and in Discussion the acronyms for brain areas start to fly thick and fast and the general reader like myself has a difficult time following. For example, Figure 1 shows lateral caudal nidopallium (NCL) (without definition in Figure 1 legend), which is not mentioned until the Discussion but ignores LMAN (first mentioned with no definition or explanation). RA is mentioned early (first mentioned without definition; also not defined in Figure 1 legend) and often, but its supposed function and anatomic definition is not given. I suggest that a more functional figure would aid discussion; one that includes all the brain areas mentioned (including, e.g., LMAN) with some at least minimal explanation of supposed function and clear anatomic definitions. Think of the general reader.

3) The writing is succinct but the paper is not always easy to follow and could benefit from more explanation for the general reader particularly in the Discussion.

Reviewer #2:

This is a very carefully executed study identifying a new pathway by which higher-order auditory information can be relayed from “cortical” auditory areas to dopamine reward centers in the midbrain. Identification of this pathway is of fundamental importance not only for our understanding of vocal learning in songbirds but more generally for setting up a framework for understanding how the dopamine system might be involved in the evaluating self-performance, a question that surprisingly has received little attention to date. The significance of this study lies not only in the identification of this new pathway, but more importantly in the evidence they provide that this pathway carries information from higher-order auditory neurons about auditory feedback error (i.e. neurons in AIV only respond to auditory feedback when feedback is perturbed) and that interruption of this pathway prevents error based vocal learning. Suggestion that this pathway might exist has been hinted at over the past two years from a number of different (behavioral, anatomical and physiological) studies but its existence remained elusive. This careful and extremely thorough study provides a significant step forward in the study of vocal learning and reinforcement learning more generally.

This study can roughly be divided into three parts:

Part I: The authors use targeted injections of retrograde and anterograde tracers to identify a major area in the arcopallium (just ventral of RA, which they call AIV (ventral intermediate arcopallium) that projects directly to the VTA. The authors then show that the AIV receives many major projections from higher-order auditory forebrain areas such as CM, L1 and HVC-shelf. Of significance is that one of these areas, CM, has been suggested to respond selectively to auditory feedback error.

Part II: To test the role this pathway might play in processing auditory feedback error, and to settle potential controversies in the field regarding an area (Ad) which lies adjacent to the AIV, the authors then perform a number of careful targeted lesions to both the AIV and Ad and test the effect of these lesions on song performance and learning. The authors show that lesions of AIV (but not Ad) in juvenile birds prevents vocal imitation of the tutor song even though birds are able to sing songs that have some degree of stereotypy. Lesions do not appear to affect adult song but are able to delay song degradation caused by deafening, a mechanism that is known to require the basal-ganglia forebrain pathway.

Part III: To evaluate the type of information that might be carried by this pathway, the authors use (mostly) antidromic stimulation to record from VTA-projecting neurons in AIV. These neurons show little responsiveness during normal singing and respond weakly, and with relatively long latencies (∼100 msec), to passively presented white noise bursts. However, when short noise bursts are presented during singing (therefore introducing an error in the feedback signal), neurons respond reliably with a short latency response. This response is specific to noise presented during singing and is not mimicked by presentation of noise during passive presentation of the bird's own song.

In general, I find this study compelling and very important and I only have one major concern:

One of the assumptions in this study is that the CM → AIV → VTA pathway “is computing or transmitting a fast online signal to VTA/SNc that potentially carries information about recent song performance”. As such each instance of song where feedback does not match motor intent, there should be a neural response in the neurons. From the data presented in Figure 5, this seems to be the case but it is difficult to be sure. This is an important question, because an alternate hypothesis is that this pathway computes the accumulation of feedback error if it occurs in a consistent manner rather than occurring online. If the neurons are responding to an accumulation of error, then one might expect responses to white noise bursts to change (increase of decrease) across trials. On the other hand, if this pathway is truly processing/monitoring error online, then responses should be similar independent of trial number. To address these potentially competing hypotheses, it would be useful for the authors to quantify responses in a slightly different ways to resolve these issues. It is likely that such analysis can be done on the data they already have.

Reviewer #3:

Mandelblat-Cerf et al. (MC et al.) investigate the connections and function of a 'cortical' area in the songbird brain that sits near the song motor nucleus 'RA'. The authors confirm and extend previous findings (especially from Gale and Perkel, and Bottjer) that neurons in this region send projections to the VTA and Substantia Nigra, and that they receive inputs from other 'cortical' structures, including auditory regions. They define the region that projects to the VTA/SN as 'AIV' and go on to demonstrate that lesions of AIV in juvenile birds cause development of abnormal song as evaluated at 90 d of age, and lesions in adults reduce deafening induced song plasticity. Additionally, they show that neurons in AIV exhibit response properties to feedback perturbation that suggest a possible role in registering vocal errors and contributing to song learning and plasticity. Overall, there is an impressive amount of data in this study that is consistent with the possibility that this brain region plays an important role in feedback guided song learning and plasticity. The authors note that the anatomically and physiologically defined outputs from AIV to the VTA/SN additionally raise the possibility that such learning depends on descending auditory evaluation signals that may reach the song system via brainstem circuitry. The study adds to our understanding of the neural substrates that contribute to vocal learning in the songbird and thereby sheds light on more general mechanisms of feedback evaluation and learning.

1) One major concern about the current study is how the results relate to previous work (Bottjer and Altenau, *Nature Neuroscience*, 2010) that assessed the effects of lesions in an adjacent/overlapping brain region. The B&A study performed lesions in a very similar brain region, that they termed 'Ad', and reported what seem to be very similar behavioral results. While CM et al. carry out some control lesions in an area that they term 'Ad' (and do not see behavioral effects reported by B&A) it is unclear whether these lesions correspond to the 'Ad' of B&A. Broadly, I found it very difficult to figure out whether CM et al. and B&A are examining the same brain regions (and calling them different things) versus examining different brain regions. This is a key issue that needs to be more clearly addressed and discussed.

2) An additional and related area of concern is that of the specificity of the chemical lesions used in the study, and specifically what damage is responsible for the observed behavioral deficits. The AIV as defined by the authors has multiple distinct regions (with different connectivity) some of which are spatially adjacent to or overlapping not only with the region termed 'Ad' by B&A, but also the 'RA cup' region and RA itself. The complex and tight packing of multiple brain regions around RA make it difficult to assess from the data that are presented what are the subdivisions of AIV, where are its boundaries, and whether the effects of lesions should be attributed specifically to damage to one or more components of AIV or to damage of other structures in the vicinity. These concerns are elaborated further below.

B&A examined the consequences of 'Ad' lesions placed lateral and ventral to RA on song learning that appear to significantly overlap with the locations of 'AIV' lesions in the current study. Also, the current study and the B&A study appear to report similar effects of these lesions (though see more questions on this important point below). B&A termed the region that they lesioned 'Ad' based on prior anatomical studies that identified several afferent and efferent projections to this region. These included projections from LMAN shell and from NCL, and included projections to VTA. Because MC et al. define AIV as the VTA/SN projecting regions surrounding RA, it would seem that the AIV as so defined includes Ad? One portion of AIV that is lesioned by MC et al. also receives projections from NC and sends projections to VTA. Hence, this portion of AIV (which appears to make up a significant part of its bulk), shares some major defining characteristics with the similarly located Ad. I presume that this portion of AIV might be distinguished (as far as anatomy is concerned) by identifying a region from which LMAN-shell afferents are excluded? I am not sure whether the authors have examined this possibility, or can otherwise clarify this issue. However, the anatomy components of the study (especially as they relate to lesion locations and locations of neural recordings) require more clarification before the relationship between the current study and previous work can be understood and evaluated.

3) The lesion coordinates for 'AIV' in MC et al. appear to overlap significantly with the lesion coordinates for 'Ad' in B&A. For example, both studies include lesions that are several hundred microns lateral to the lateral most edge of RA, approximately at the depth of the center of RA (as well as more ventrally), and at approximately the R-C center of RA (as well as more rostrally and caudally). With the listed coordinates for both studies it would seem plausible if not likely that the two studies have lesioned overlapping brain regions. Additionally, MC et al. carry out separate control 'Ad' lesions (intended to test whether the reported effects could be due to lesions of Ad rather than AIV. However, the R-C and M-L locations of these Ad lesions (in MC et al.) fall into this same region lateral to RA. The D-V locations of these lesions are not provided, but instead the authors note that Ad lesions were made at depths that corresponded to the electrophysiologically defined boundaries of Ad. The authors should provide the D-V locations of Ad lesions, and some validation of the physiological criteria for defining the boundaries of Ad. I suppose that the authors' intent with their Ad lesions was to show that lesions restricted to the region lateral to RA (that may have been lesioned in both the 'AIV' lesions of CM et al. and the 'Ad' lesions of B&A) are insufficient to elicit the behavioral deficits associated with the 'AIV' lesions. However, because the authors use a lower concentration (and volume) of NMA for their Ad lesions than for their AIV lesions (1% versus 2%) this argument needs to be bolstered with some careful analysis of histology to establish exactly what was destroyed in each experiment. If I understand, the authors' argument is that lesions to 'AIV' that exclude the region lateral to RA (where they made ineffective 'Ad lesions') would be effective for disrupting learning. Are there examples with this pattern of damage? More broadly, the authors need to show us more of the histology that is associated with their lesions of AIV and of Ad and provide more detail on the criteria used for distinguishing between these regions, and also provide some more help in determining the relationship between the current study and prior work.

4) Other concerns about the lesions include the possibility of damage to RA. The authors note that 2 birds were excluded for which RA was substantially damaged by AIV lesions. However, other birds that had lesions of up to 5% of RA were included as 'AIV lesions'. Could this have contributed to behavioral deficits? Again, the close packing of structures in the vicinity of RA, and the difficulty of defining the borders of chemical lesions, raises concerns over the degree to which the observed effects can be attributed specifically to lesions of the anatomically defined AIV.

5) Effects of lesions on amount of singing. The authors dismiss the possibility that the poor learning in AIV lesioned birds could have occurred because lesions reduced the amount of singing by juvenile birds during the learning period: ”Importantly, the impaired vocal imitation in AIV-lesioned birds cannot be attributed to the amount of vocal practice. We compared the amount of singing in the period from surgery up to 90dPh for AIV-lesioned birds versus AD-lesioned controls. No difference was observed...” I believe that most people would read this as an indication that AIV lesions did not reduce the amount of singing. However, the supplementary material appears to show that birds with AIV lesions sang significantly less than intact control birds (though this is not reported, it looks like there is about a 30% reduction, on average, in amount of singing in AIV lesioned birds). If this is the case, the authors should note this important observation in the main text and address how this influences interpretation of lesions results with respect to deficits of learning and song maturity as assessed at 90d of age. Given the large amount of individual variation.

6) Effects of lesions on learning: I found it difficult to get a sense of how song learning was affected in the AIV lesion birds. The authors devise a 'tutor imitation score' that is a single number intended to capture how closely the juvenile songs (at 90d of age) match the tutor song (in terms of both the copying of individual notes and of the sequencing of notes). Because this measure has not been used in previous studies of birdsong, it is difficult to interpret. The authors provide as reference the imitation score for unrelated adult songs, and suggest that AIV lesion birds are almost as dissimilar to the tutor song as unrelated adult birds' songs are to each other. However, there are many ways in which the quality of match between experimental and tutor songs may differ and it is important in reporting results and for purposes of comparing with previous studies for the authors to figure out how to provide more clarity about how the AIV songs differ from the tutor songs. For example, deafening, lesions of LMAN, lesions of X and lesions of Ad (in B&A) in juvenile birds all appear to disrupt imitation, but in different ways. CM et al., suggest that the nature of song disruption caused by their AIV lesions may be similar to the effects of X-lesions in juvenile birds (as reported by Scharff and Nottebohm). However, the single example that is shown (Figure 3) does not look particularly like the examples of X-lesion birds in Scharff's study.

This may be because this example comes from an AIV lesion bird that is less disrupted than most. From supplementary data it appears that this AIV lesion bird has an imitation score of about .21, while its control sibling has an imitation score of about .24. Hence, this example of disrupted learning from an AIV lesion bird shows very modest disruption (this AIV lesion bird has a better tutor imitation score than the majority of control birds!). It would be helpful if the authors presented spectrograms from more birds to give a clearer qualitative sense of what is going in, and some context for better understanding what different values of the imitation score (and other new measures introduced here) mean. This should include examples of good and bad imitation from controls and from AIV lesion (and perhaps also Ad lesion) birds. For example, do the control birds that have imitation scores around 0.1 and below look as dissimilar to the tutor song as do the unrelated adults? Are these control birds disrupted in a qualitatively similar manner to the AIV lesion birds that also have scores in that range? Regardless of what measures the authors use, they need to more clearly convey to the reader the extent to which the effects of AIV lesions parallel the effects of X-lesions (as reported by Scharff) and/or Ad lesions (as reported by B&A).

---

## [Author Response]

*Overview*:

*In this manuscript the authors present careful anatomical, electrophysiological, and behavioral analyses in songbirds of a brain area they term ventral intermediate arcopallium (AIV), which receives auditory cortical input and has output to the dopaminergic VTA/SNc. They further show that AIV neurons that are demonstrated to project to VTA by antidromic activation respond to disruptive auditory feedback during singing but not during the playback of the bird's own song or the bird's own song with disruptive auditory feedback by in vivo single unit (and multiunit) recording in awake unrestrained birds during singing and quiescence. They also show that lesions of the AIV leads to deficits in vocal learning but do not alter song in mature birds. The data set presented is extensive and the analyses quantitative. The AIV neurons described provide an opening to the dopaminergic reward system that promises in future to help elucidate how auditory feedback can alter vocal output: a significant problem in song birds and humans. This analysis will generate wide interest in the songbird and sensory motor integration communities*.

*Concerns*:

*There are several concerns in the reviews by the BRE and the external reviewers that are provided to the authors to help them in revision. To focus this revision and remove any apparent conflict in the reviews, a consensus has formed about three areas*.

Anatomy:

*The reviewers are aware that as work has proceeded in the songbird, focus has shifted from distinct nuclei like RA or X to more diffusely organized areas like the area described here as AIV. AIV must be as precisely delimited as is possible and the authors should be at pains to do so. The anatomy presented in support of AIV must be discussed in the context of past findings, particularly*
[8]
*(Parallel pathways for vocal learning in basal ganglia of songbirds. Nature Neuroscience 13, 153-155.), even if this makes sections of the Discussions more for birdsong experts*.

We have very precisely defined the boundaries of AIV in terms of the extent of neurons retrogradely-labeled from VTA/SNc. While the extent of AIV is quite reliable from bird-to-bird relative to nucleus RA, all of our anatomical work identifying the inputs to AIV is based on retrograde-labeling of neurons from VTA/SNc.

Although we have found no clear cytoarchitectonic features (visible without retrograde labeling) by which to identify the boundaries of AIV, we believe the retrograde labeling provides a practical approach to defining AIV. Fortunately, injections into VTA/SNc are relatively straightforward, as the thalamic nucleus Ovoidalis can be used as a local reference point.

As described below, we have substantially expanded our description of the relation between AIV and Ad.

*It would also serve the community if some other nearby areas could be differentiated from or included in AIV e.g. RA cup. This is not to set previous work as some anatomical standard but rather to motivate further clarification that will be healthy for the field*.

Definitions of RA cup rely on two types of information: 1) Afferents to RAcup from primary auditory fields L1, L3, and auditory area HVC shelf. 2) Efferents from RAcup to thalamic and midbrain areas Ovoidalis shell and MLD. We have a large amount of data now, from double labeling studies, suggesting that all of the regions in ventral arcopallium identified by these earlier studies correspond to different partially-overlapping or even non-overlapping regions. Thus, in our view now, there is no such thing as ‘RA cup’ per se, but rather a complex heterogeneous pattern of auditory-related projections into and out of ventral arcopallium.

In our study, we have shown the relation between AIV neurons and projections from HVC-shelf and L1, two of the defining features of RAcup, and have identified a new input to the ventral arcopallium from auditory area CM.

An effort to further elucidate the relation between AIV and RA cup is ongoing. We are now focusing on the efferents from the ventral arcopallium to Ov-shell and MLD. We fully agree with the reviewers that this needs to be done, and it will be done. However, this is a substantial effort, with another set of 3-4 anatomical studies involving double or triple labeling. We feel that this new work is beyond the scope of the present paper.

*Not only should the authors specify exactly what the boundaries of AIV are but they should thoroughly discuss how it is related to Ad. They need to state specifically why they think they have lesioned one and not the other and how the AIV vs. Ad lesions lead to different or similar behavioral results. They should try to explain discrepancies in their data and that from*
[8].

We have added additional anatomical findings that more clearly elucidate the relation between AD and AIV. Our data suggest that the effects of AD lesions reported by Bottjer and Altenau were the result of unintended lesions of AIV in that study. We have now extensively expanded the discussion of the earlier findings on Ad, and specifically addressed an explanation for the discrepancies.

*Many of the projections described to AIV have been previously described, at least in part. The projection from CM to AIV is novel and potentially most functionally significant for the physiological responses of AIV to VTA projecting neurons; it should be highlighted*.

We are delighted that the reviewer appreciates the novelty of the finding that CM projects to AIV. The Discussion already includes an extensive paragraph on the anatomical and function significance of this finding. However, we have added a sentence to the Results section pointing out the novelty of this result.

*Behaviour*:

*The authors should substantially increase the presentation of their primary data on song that is encompassed in their imitation score. There are no figure limitations in eLife, so it is possible to present more sonograms so that other workers can more easily judge how song is affected or not affected by a lesion. In addition to changes in imitation score, they should give a qualitative description of how song is affected by their lesions. They should then thoroughly discuss how the results are similar to or different than Ad and X lesions by others*.

In order to incorporate more sonograms, we have split the old Figure 3 into two new figures. The first of these, now Figure 4, shows four sets of sonograms from lesioned birds. (Two of these have non-lesioned siblings from the same tutor.) The second of the new figures, now Figure 5, shows the quantitative analysis of song imitation.

We have also added additional qualitative descriptions of how song is affected, specifically the apparent high incidence of atypical non-song-like vocal elements.

However, the incidence of such vocal elements is difficult to quantify, and we would emphasize that we have been justifiably cautious about making such statements.

*Physiology*:

*There are a number of specific comments by Reviewer #2 that should be addressed*.

See below for response to these comments.

*The Abstract is written quite conservatively and is highly appropriate; it might be best to adjust the title so that it more matches the conservative tone of Abstract*.

The title has been changed: ‘A role for descending auditory cortical projections in songbird vocal learning’

*Other issues to address*:

*Reviewer #1*:

*1) The work is carefully done with detailed analysis and the figures appropriately illustrate the data. Nevertheless the figures are very complicated and could benefit from more detailed legends and perhaps better labeling. A tremendous amount of data is presented in supplementary figures. eLife discourages use of supplementary data and rather includes all important data in the text*.

We have indeed presented a lot of data in the supplementary information and have now made an effort to move some of this to the main figures. We feel that all the essential data are now included in the main figures, and that the data presented in supplementary information only support those presented in the main figures, rather than representing significant findings on their own. We feel that moving more of these results or analysis to the main findings would be unduly burdensome to the reader.

At the same time, we feel that some readers will find the supplementary data and analysis extremely helpful, and would very much like to keep these as part of the paper.

*The authors should ruthlessly select the relevant data from the supplementary data and present those integrated into the text. For example the lower panels in*
Figure 5*—figure supplement 5 could be eliminated. I defer to the expert reviewers and the authors on what is essential, however*.

We have moved parts of the old Supplementary Figure 2 to the main text as the new Figure 3. As requested, we have eliminated the lower panels of Figure 5—figure supplement 5 (now Figure 7—figure supplement 1)

*2) A better general explanation (and figure) of the system than afforded in the Introduction, Discussion and by*
Figure 1
*would greatly help the general reader especially in Discussion. I fully understand why*
Figure 1
*is simplified to explain the anatomical experiments, but by the time we move beyond these experiments through Results and in Discussion the acronyms for brain areas start to fly thick and fast and the general reader like myself has a difficult time following. For example,*
Figure 1
*shows lateral caudal nidopallium (NCL) (without definition in*
Figure 1
*legend), which is not mentioned until the Discussion but ignores LMAN (first mentioned with no definition or explanation). RA is mentioned early (first mentioned without definition; also not defined in*
Figure 1
*legend) and often, but its supposed function and anatomic definition is not given. I suggest that a more functional figure would aid discussion; one that includes all the brain areas mentioned (including, e.g., LMAN) with some at least minimal explanation of supposed function and clear anatomic definitions. Think of the general reader*.

We have substantially augmented the explanation of the various song components. We have added a new panel to Figure 1 showing the classical song-control brain areas, and have been more generous with the explanations in the Discussion. Finally, the original Figure 1 erroneously labeled NC as NCL, which explains why this did not appear until the Discussion. We thank the reviewers for catching this, and apologize for the confusion it must have caused.

3) The writing is succinct but the paper is not always easy to follow and could benefit from more explanation for the general reader particularly in the Discussion.

*Reviewer #2*:

*In general, I find this study compelling and very important and I only have one major concern*:

*One of the assumptions in this study is that the CM → AIV → VTA pathway “is computing or transmitting a fast online signal to VTA/SNc that potentially carries information about recent song performance”. As such each instance of song where feedback does not match motor intent, there should be a neural response in the neurons. From the data presented in*
Figure 5*, this seems to be the case but it is difficult to be sure. This is an important question, because an alternate hypothesis is that this pathway computes the accumulation of feedback error if it occurs in a consistent manner rather than occurring online. If the neurons are responding to an accumulation of error, then one might expect responses to white noise bursts to change (increase of decrease) across trials. On the other hand, if this pathway is truly processing/monitoring error online, then responses should be similar independent of trial number. To address these potentially competing hypotheses, it would be useful for the authors to quantify responses in a slightly different ways to resolve these issues. It is likely that such analysis can be done on the data they already have*.

We thank the reviewer for this important suggestion. We have analyzed the data as suggested: i.e. look to determine if the error-related responses increase or decrease over the period of exposure to noise bursts. Our analysis shows that the responses do not show a significant trend over time (no significant correlation of response with trial number, F-statistics p>0.11 for all neurons). Furthermore, the responses are statistically indistinguishable in the first 20 trials and last 20 trials. (Ranksum test, p>0.15)

*Reviewer #3*:

*1) One major concern about the current study is how the results relate to previous work (Bottjer and Altenau, Nature Neuroscience, 2010) that assessed the effects of lesions in an adjacent/overlapping brain region. The B&A study performed lesions in a very similar brain region, that they termed 'Ad', and reported what seem to be very similar behavioral results. While CM et al. carry out some control lesions in an area that they term 'Ad' (and do not see behavioral effects reported by B&A) it is unclear whether these lesions correspond to the 'Ad' of B&A. Broadly, I found it very difficult to figure out whether CM et al. and B&A are examining the same brain regions (and calling them different things) versus examining different brain regions. This is a key issue that needs to be more clearly addressed and discussed*.

We have carried out extensive anatomical characterization of the relation between AIV and Ad. AIV is defined as a region in the ventral arcopallium containing neurons projecting to VTA and SNc. Ad is defined as a region receiving afferents from the shell-region around LMAN (LMANshell). The distinction between these areas is most clearly visible in coronal slices (Figures 1 and 5). To show this relation more clearly in this paper, we have carried out additional experiments using double labeling (n=3 birds, 6 hemispheres). Anterograde tracer (dextran-Alexa488) was injected into LMANshell and retrograde tracer (CTB-Alext647) was injected into VTA/SNc. Anterogradely-labelled fibers were visible in a band lateral to RA, consistent with earlier descriptions of Ad. Neurons retrogradely-labelled from VTA/SNc were visible in a ‘wedge’ ventral and medial to Ad. Very few neurons retrogradely labeled from VTA/SNc were seen within AD.

Furthermore, injections of retrograde tracer into AIV revealed no evidence of labeled neurons in LMAN-shell. Thus, the anterograde and retrograde tracing experiments suggest that Ad and AIV are distinct circuits with non-overlapping patterns of afferents and efferents.

*2) An additional and related area of concern is that of the specificity of the chemical lesions used in the study, and specifically what damage is responsible for the observed behavioral deficits. The AIV as defined by the authors has multiple distinct regions (with different connectivity) some of which are spatially adjacent to or overlapping not only with the region termed 'Ad' by B&A, but also the 'RA cup' region and RA itself. The complex and tight packing of multiple brain regions around RA make it difficult to assess from the data that are presented what are the subdivisions of AIV, where are its boundaries, and whether the effects of lesions should be attributed specifically to damage to one or more components of AIV or to damage of other structures in the vicinity. These concerns are elaborated further below*.

We have carried out preliminary tracing experiments using double-labeling to characterize the relation between AIV and RA-cup. Our results suggest that the various afferents and efferents previously used to define RA cup do not correspond to a single area, but are rather a set of partially-overlapping inter-related areas. Some of these areas overlap with AIV, as we have defined it, and some do not. Our efforts to elucidate the relation between AIV and RAcup are ongoing, and we believe that the present manuscript will provide strong motivation for future work in this area.

We emphasize that our definition of AIV is very precise: ‘The region of arcopallium retrogradely labeled by injection of retrograde tracer into VTA/SNc.’

*B&A examined the consequences of 'Ad' lesions placed lateral and ventral to RA on song learning that appear to significantly overlap with the locations of 'AIV' lesions in the current study. Also, the current study and the B&A study appear to report similar effects of these lesions (though see more questions on this important point below). B&A termed the region that they lesioned 'Ad' based on prior anatomical studies that identified several afferent and efferent projections to this region. These included projections from LMAN shell and from NCL, and included projections to VTA. Because MC et al. define AIV as the VTA/SN projecting regions surrounding RA, it would seem that the AIV as so defined includes Ad? One portion of AIV that is lesioned by MC et al. also receives projections from NC and sends projections to VTA. Hence, this portion of AIV (which appears to make up a significant part of its bulk), shares some major defining characteristics with the similarly located Ad. I presume that this portion of AIV might be distinguished (as far as anatomy is concerned) by identifying a region from which LMAN-shell afferents are excluded? I am not sure whether the authors have examined this possibility, or can otherwise clarify this issue. However, the anatomy components of the study (especially as they relate to lesion locations and locations of neural recordings) require more clarification before the relationship between the current study and previous work can be understood and evaluated*.

The reviewer is correct that the anatomical relation between AIV and Ad must be described before it is possible to understand our findings in comparison to the Altenau and Bottjer. We have done this very carefully. Afferents from LMAN-shell overlap minimally with neurons retrogradely-labeled from VTA/SNc. This might suggest that the claim that Ad projects to VTA could be an artifact of leakage of tracer from Ad into neighboring AIV.

As noted above, we have re-examined our earlier histological material in which retrograde tracers were injected into AIV. We found no evidence for retrogradely-labeled neurons in the vicinity of LMAN (LMAN-shell). Thus both the anterograde and retrograde tracing experiments strongly support the conclusion that AIV and Ad are anatomically distinct circuits, with little or no overlap in either afferent or efferent connections.

*3) The lesion coordinates for 'AIV' in MC et al. appear to overlap significantly with the lesion coordinates for 'Ad' in B&A. For example, both studies include lesions that are several hundred microns lateral to the lateral most edge of RA, approximately at the depth of the center of RA (as well as more ventrally), and at approximately the R-C center of RA (as well as more rostrally and caudally). With the listed coordinates for both studies it would seem plausible if not likely that the two studies have lesioned overlapping brain regions. Additionally, MC et al. carry out separate control 'Ad' lesions (intended to test whether the reported effects could be due to lesions of Ad rather than AIV. However, the R-C and M-L locations of these Ad lesions (in MC et al.) fall into this same region lateral to RA. The D-V locations of these lesions are not provided, but instead the authors note that Ad lesions were made at depths that corresponded to the electrophysiologically defined boundaries of Ad. The authors should provide the D-V locations of Ad lesions, and some validation of the physiological criteria for defining the boundaries of Ad. I suppose that the authors' intent with their Ad lesions was to show that lesions restricted to the region lateral to RA (that may have been lesioned in both the 'AIV' lesions of CM et al. and the 'Ad' lesions of B&A) are insufficient to elicit the behavioral deficits associated with the 'AIV' lesions. However, because the authors use a lower concentration (and volume) of NMA for their Ad lesions than for their AIV lesions (1% versus 2%) this argument needs to be bolstered with some careful analysis of histology to establish exactly what was destroyed in each experiment*.

The reviewer is presumably concerned here with the possibility that the Ad lesions may have been too small to have an effect on vocal learning. We point out that the protocol for lesioning Ad was validated in three different birds. The pattern of lesion injections was targeted by first electrophysiologically mapping Ad, in each lesioned bird, based on its distinctive bursting activity (similar to that observed in RA). The lesions were confirmed histologically after the birds reached adulthood.

Finally, in an attempt to reproduce the results of Bottjer and Altenau, we carried out an additional set of experiments in which we made complete lesions of Ad. These birds exhibited severe akinesia and immobility, such that the experiments had to be terminated and song imitation could not be assessed. These findings are consistent with the hypothesis (Jarvis, 2008) that Ad is involved in locomotion and other motor behaviors, rather than song learning.

Given the reviewers’ concern about the relation between our findings and those of [8], we have now reported the results of these larger lesions.

*If I understand, the authors' argument is that lesions to 'AIV' that exclude the region lateral to RA (where they made ineffective 'Ad lesions') would be effective for disrupting learning. Are there examples with this pattern of damage? More broadly, the authors need to show us more of the histology that is associated with their lesions of AIV and of Ad and provide more detail on the criteria used for distinguishing between these regions, and also provide some more help in determining the relationship between the current study and prior work*.

We assessed our lesions of AIV in saggitally-sectioned tissue, because it is much easier to identify the boundaries of AIV in these sections, where RA serves as a clearly-visible landmark. However, it is not possible to assess the impact of these lesions on Ad in sagittal sections, which is lateral to RA. Therefore, we are not able to produce the evaluation, requested by the reviewer, of the impact of our ‘AIV lesions’ on Ad in the experimental birds.

However, we carried out a careful calibration study, prior to our lesion experiments, showing that our AIV lesion protocol had minimal impact on AD, as assessed in coronally-sliced tissue in which Ad was labeled by anterograde tracing from LMAN-shell. Furthermore, our Ad lesion control experiments were specifically designed to address this issue. Again, we found no effect of Ad lesions on vocal imitation.

*4) Other concerns about the lesions include the possibility of damage to RA. The authors note that 2 birds were excluded for which RA was substantially damaged by AIV lesions. However, other birds that had lesions of up to 5% of RA were included as 'AIV lesions'. Could this have contributed to behavioral deficits? Again, the close packing of structures in the vicinity of RA, and the difficulty of defining the borders of chemical lesions, raises concerns over the degree to which the observed effects can be attributed specifically to lesions of the anatomically defined AIV*.

In only three of the birds included as ‘AIV lesions’ did the lesions have any impact on RA. In these cases, a thin rim of RA was impacted, and only in a few of the 100um sections was this damage observed. Furthermore, in these cases only one hemisphere was involved. We have reanalyzed the behavioral results with these birds removed from the dataset, and our findings are still highly significant (p=0.0084).

*5) Effects of lesions on amount of singing. The authors dismiss the possibility that the poor learning in AIV lesioned birds could have occurred because lesions reduced the amount of singing by juvenile birds during the learning period: “Importantly, the impaired vocal imitation in AIV-lesioned birds cannot be attributed to the amount of vocal practice. We compared the amount of singing in the period from surgery up to 90dPh for AIV-lesioned birds versus AD-lesioned controls. No difference was observed...” I believe that most people would read this as an indication that AIV lesions did not reduce the amount of singing. However, the supplementary material appears to show that birds with AIV lesions sang significantly less than intact control birds (though this is not reported, it looks like there is about a 30% reduction, on average, in amount of singing in AIV lesioned birds). If this is the case, the authors should note this important observation in the main text and address how this influences interpretation of lesions results with respect to deficits of learning and song maturity as assessed at 90d of age. Given the large amount of individual variation*.

The reviewer is correct that the AIV lesions and the AD lesions had an effect on the total amount of singing compared to unmanipulated birds. This is not surprising, given that both of groups of lesioned birds underwent a significant surgical manipulation, while the unmanipulated birds had no surgery. However, we found that the AD lesioned birds showed a reduction in the amount of singing, but no deficits in song learning. Furthermore, we found no correlation between the amount of singing and the degree of song imitation. We believe we have properly controlled for any effect of amount of singing on song imitation. These points have been raised in the results section.

*6) Effects of lesions on learning: I found it difficult to get a sense of how song learning was affected in the AIV lesion birds. The authors devise a 'tutor imitation score' that is a single number intended to capture how closely the juvenile songs (at 90d of age) match the tutor song (in terms of both the copying of individual notes and of the sequencing of notes). Because this measure has not been used in previous studies of birdsong, it is difficult to interpret. The authors provide as reference the imitation score for unrelated adult songs, and suggest that AIV lesion birds are almost as dissimilar to the tutor song as unrelated adult birds' songs are to each other. However, there are many ways in which the quality of match between experimental and tutor songs may differ and it is important in reporting results and for purposes of comparing with previous studies for the authors to figure out how to provide more clarity about how the AIV songs differ from the tutor songs*.

Song imitation is a highly variable process: different birds normally copy different aspects of the tutor song. Furthermore, different pupil birds have different degrees of acoustic and sequence stereotypy. This makes it very difficult to convey, in a few examples, the precise nature of the imitation. We have selected two aspects of song imitation on which to focus our quantification (acoustic similarity and sequence similarity with the tutor song). We have developed a largely automated procedure for quantifying these measures, and a paper describing this procedure has now been published in PLOS One. The Matlab code for implementing this algorithm was made publicly available.

*For example, deafening, lesions of LMAN, lesions of X and lesions of Ad (in B&A) in juvenile birds all appear to disrupt imitation, but in different ways. CM et al., suggest that the nature of song disruption caused by their AIV lesions may be similar to the effects of X-lesions in juvenile birds (as reported by Scharff and Nottebohm). However, the single example that is shown (*Figure 3*) does not look particularly like the examples of X-lesion birds in Scharff's study*.

*This may be because this example comes from an AIV lesion bird that is less disrupted than most. From supplementary data it appears that this AIV lesion bird has an imitation score of about .21, while its control sibling has an imitation score of about .24. Hence, this example of disrupted learning from an AIV lesion bird shows very modest disruption (this AIV lesion bird has a better tutor imitation score than the majority of control birds!). It would be helpful if the authors presented spectrograms from more birds to give a clearer qualitative sense of what is going in, and some context for better understanding what different values of the imitation score (and other new measures introduced here) mean. This should include examples of good and bad imitation from controls and from AIV lesion (and perhaps also Ad lesion) birds. For example, do the control birds that have imitation scores around 0.1 and below look as dissimilar to the tutor song as do the unrelated adults? Are these control birds disrupted in a qualitatively similar manner to the AIV lesion birds that also have scores in that range? Regardless of what measures the authors use, they need to more clearly convey to the reader the extent to which the effects of AIV lesions parallel the effects of X-lesions (as reported by Scharff) and/or Ad lesions (as reported by B&A)*.

As requested, we have added additional examples of song spectrograms in AIV-lesioned birds. However, we feel that the quantitative procedure we have developed is much more informative, as it captures the quality of many hundreds of song renditions rather than just one or two examples.

As the reviewer suggests, we have also examined the songs of the two low-performing control birds. Their songs appear to be quite dissimilar to their tutor songs, and qualitatively speaking, are as different from their tutors as unrelated adults. The songs of these controls birds differ from their tutors in a manner that is not inconsistent with the way AIV-lesioned birds exhibited poor imitation performance, including a lack of acoustic and sequence similarity and high variability. We should note here that the imitation of tutor song by zebra finches can be variable, particularly in birds that are isolated from their tutors at an early age, as was done in our experiments. We don’t feel that adding a few examples of song spectrograms from good and bad imitators in each experimental group will clarify our findings. Our claims are based on an automated quantitative assessment of song imitation and variability.

Our statement that AIV lesions resemble lesions of Area X was based on the fact that these lesions in juvenile birds result in poor imitation and higher song variability, while the same lesions in adult birds have little effect on learned song. We do not wish to claim that the AIV lesions produced the exact same effect as X lesions. There is no reason to expect such a precise correspondence of outcomes from lesions in these different brain areas. We have softened the wording in the relevant paragraph of the Discussion.

The reviewer raises an interesting point about the relation between the effects of AIV lesions and those of Ad lesions. The Bottjer and Altenau study claim that lesions of Ad have no immediate effect on song, produce poor imitation and increase song variability. These findings are consistent with our suggestion that the effects of lesions targeted by Ad reported by Bottjer and Altenau are the result of an unintended lesion of AIV.